# Shared EEG correlates between non-REM parasomnia experiences and dreams

Jacinthe Cataldi[1,2], Aurélie M. Stephan[1,2,3], José Haba-Rubio[1] & Francesca Siclari [1,2,3] ✉

Sleepwalking and related parasomnias result from incomplete awakenings out of non-rapid eye movement sleep. Behavioral episodes can occur without consciousness or recollection, or in relation to dream-like experiences. To understand what accounts for these differences in consciousness and recall, here we recorded parasomnia episodes with high-density electro-encephalography (EEG) and interviewed participants immediately afterward about their experiences. Compared to reports of no experience (19%), reports of conscious experience (56%) were preceded by high-amplitude EEG slow waves in anterior cortical regions and activation of posterior cortical regions, similar to previously described EEG correlates of dreaming. Recall of the content of the experience (56%), compared to no recall (25%), was associated with higher EEG activation in the right medial temporal region before movement onset. Our work suggests that the EEG correlates of parasomnia experiences are similar to those reported for dreams and may thus reflect core physiological processes involved in sleep consciousness.

Sleepwalking (somnambulism) and related parasomnias are enigmatic conditions characterized by sudden but partial awakenings out of non-rapid eye movement (NREM) sleep[1], during which affected individuals may interact with their environment in an altered state of consciousness[2–4]. Behaviors during parasomnia episodes can be short and simple, such as sitting up and talking, or more complex, like sustaining a conversation, leaving the bed, or manipulating objects. In extreme cases, sleepwalkers have been reported to drive, commit sexual assault, homicide, and pseudo-suicide, resulting in personal tragedies and legal dilemmas[5–7]. Although rare, these instances invariably raise the question of whether sleepwalkers are conscious, and if yes, what they experience during parasomnia episodes.

Anecdotal reports describing dream-like experiences during sleepwalking date back to the eighteenth century[5], but were largely dismissed in the 1960s, when the first electroencephalographic (EEG) recordings of somnambulistic episodes showed that this condition occurred out of slow wave sleep[8,9], a sleep stage that was considered largely dreamless and in stark opposition to the recently discovered rapid eye movement (REM) sleep[10,11]. Although it was later

acknowledged that patients could report 'apparent dream recall' after parasomnia episodes[12,13], some questioned that this memory truly reflected dreaming[9], and the view that sleepwalking predominantly represented a largely unconscious, 'ambulatory automatism'[8] prevailed. In subsequent years, paralleling the growing evidence for NREM dreaming[14], dream-like experiences during NREM parasomnia episodes were increasingly reported[3,15–26]. These experiences most often consisted in threatening scenarios[15,19–21] and could encompass hallucinations[23–25], delusions[16,26], and behaviors in line with the 'dream' content. Larger studies, in which patients were asked to describe, over a lifetime, what they had experienced during NREM parasomnia episodes revealed that 66–94% of adults could report at least one experience[21,22,25], while this rate was only about a third in children[17]. There is also anecdotal evidence of NREM parasomnia experiences with minimal or partial consciousness, and with variable degrees of amnesia[6]. However, almost all of these studies relied on retrospective recall of experiences over a lifetime, and are therefore prone to recall biases, especially since it is well known that dream-related memories vanish rapidly[27]. In one of the rare studies in which conscious

[1]Center for Investigation and Research on Sleep, Lausanne University Hospital, Lausanne, Switzerland. [2]The Sense Innovation and Research Center, Lausanne and Sion, Switzerland. [3]The Netherlands Institute for Neuroscience, Amsterdam, The Netherlands. ✉e-mail: f.siclari@nin.knaw.nl

**Table 1 | Behavioral features of parasomnia episodes**

| Feature | Presence, N (%) | Absence, N (%) | No agreement, N (%) | Other reason (N) |
|---|---|---|---|---|
| Eye-opening | 60 (80%) | 4 (5%) | 2 (3%) | Not visible (4), both open/closed (9) |
| Sudden onset | 49 (65%) | 24 (32%) | 2 (3%) | – |
| Perplexity | 48 (64%) | 21 (28%) | 2 (3%) | Not visible (4) |
| Orienting behavior | 45 (60%) | 28 (37%) | 1 (1%) | Not visible (1) |
| Somniloquia | 43 (57%) | 32 (43%) | 0 | – |
| Interactions with environment | 29 (39%) | 42 (56%) | 4 (5%) | – |
| Startle-like reaction at onset | 28 (37%) | 45 (60%) | 2 (3%) | – |
| Expression of fear | 17 (23%) | 53 (71%) | 2 (3%) | Not visible (3) |
| Smiling or laughing | 3 (4%) | 66 (88%) | 2 (3%) | Not visible (4) |

Presence/absence of feature (%) refers to instances in which at least two of the three raters agreed. Note that, depending on the feature in question, more than two categories or ratings were possible (presence/absence/not visible/both present and absent within the same episode).

experiences (CEs) were assessed immediately after parasomnia episodes in patients with sleep terrors, vivid experiences were reported in 58% of cases, and vague recall in 7%[12]. Thus, the level of consciousness and amnesia associated with NREM parasomnia episodes is likely more variable than previously thought, ranging from episodes with no or minimal consciousness or recall to vivid, dream-like experiences.

Our first aim was therefore to systematically assess and quantify this variability by interviewing patients immediately after NREM parasomnia episodes about their experiences. Our second aim was to understand which brain activity changes, at the cortical level, could account for the variability in consciousness and amnesia associated with parasomnia episodes. Previous studies using nuclear imaging and intracranial EEG recordings documented patterns of wake- and sleep-like activity in different brain regions during NREM parasomnia episodes[4,28–31], but these techniques are not readily available to study a large number of episodes. It remains therefore unclear how these dissociated brain activity patterns relate to CE. Here we took advantage of high-density (hd-) EEG recordings to record parasomnia episodes[32,33]. In combination with a serial interview paradigm, this technique has previously allowed us to identify brain activity patterns that distinguish unconsciousness from dreaming in both REM and NREM sleep in healthy sleepers[34,35]. More specifically, our previous studies showed that compared to reports of unconsciousness, reports of dreaming were preceded by a regional EEG activation in parieto–occipital brain areas (grouped under the name 'posterior hot zone')[34], and in NREM sleep, by high-amplitude frontal slow waves (type I slow waves, encompassing K-complexes) that are likely related to arousal systems[35]. We hypothesized that if these EEG features reflect core physiological processes involved in sleep consciousness, they should also distinguish parasomnia episodes with and without CE.

## Results

### Patients
We included twenty-two patients with disorders of arousal [14 of the female sex, aged 26.9 ± 5.3 years (average ± SD), range 18.3–36.3 years, see supplementary material Text S1 for additional information on patients]. Twenty of these participants were included in a previous publication[36] while two participants were newly recruited. Participants underwent two high-density EEG sleep recordings: a first nighttime recording and a second daytime recording after 25 h of total sleep deprivation, during which acoustic stimuli were administered to increase the complexity and incidence of parasomnia episodes[37–39] (see "Methods" section for details). Immediately after each parasomnia episode, the examiner called the patients by name and asked about their most recent experience (based on refs. 34,40). When participants did not report a CE, they had to clarify whether they had experienced something but could not recall/report the content of the experience (CE without report of content, CEWR), or whether they had not experienced anything (no experience, NE).

### Parasomnia episodes
One hundred two potential parasomnia episodes [19 patients, 5.3 ± 3.54 episodes per patient, range 1–12] were recorded and examined by three independent raters. Of these, 75 (73.5%, 18 patients) were unanimously rated as parasomnia episodes by all three experts and were included in the analyses. Parasomnia episodes mostly consisted of short confusional arousals. None of the patients left the bed, although the setup allowed them to do so. Episodes lasted on average 28.6 s (+/−18.8 s, median 22.0 s, range 3–108 s), and occurred 149.7 min (+/−91.5) after lights off (range 8–373 min). Of the 75 episodes unanimously rated as parasomnia episodes, 48 (64.0%) occurred spontaneously and 27 (36.0%) were provoked by sounds. Thirty-seven episodes (49.3%) occurred during the baseline night, and 38 (50.7%) during the recovery night. Sixty-nine episodes (92.0%) occurred out of stage N3, and 6 (8.0%) out of stage N2.

### Behavioral features of parasomnia episodes
Most episodes had a sudden onset and were characterized by eye-opening and manifestations of surprise, including orienting behaviors (exploratory head and eye movements) and expressions of perplexity, as well as somniloquy (Table 1). Interestingly, manifestations of surprise were not only observed at the onset of provoked parasomnia episodes, but also with a similar frequency in spontaneous episodes, when no sound was played, consistent with our previous observation that these two types of episodes are behaviorally indistinguishable[36]. Indeed, a generalized linear model was not able to predict the provoked vs spontaneous nature of the behavior based on the presence of surprise features (included as a fixed factor while accounting for subject identity as a random factor, [($X^2(1) = 0.22$, $p = 0.64$), see "Methods" and Table S1 for information on statistical models]).

More complex behaviors were seen in longer episodes (>30 s) and included searching (going through the bed sheets, looking under the bed), gesticulating (trying to catch or grasp non-existent objects, military salutation), sustaining imaginary conversations (i.e., alternatively speaking and pausing, as if listening), visually scanning the environment for continued periods of time and/or pointing towards (imaginary) objects. Apparent hallucinations and/or perceptual illusions were present in 52% of episodes.

### Consciousness and parasomnia episodes
Of the 75 unanimously scored episodes, 57 (76%) were followed by an interview about CEs. Instances without interview occurred either because the experimenter failed to recognize the parasomnia episode as such during the recording, or because the patient fell asleep too quickly after the episode to answer questions. CE was reported in 56.1 % ($n = 32$, 14 patients), CEWR in 24.6% ($n = 14$, 8 patients) and NE in 19.3% ($n = 11$, 7 patients Fig. 1A, B).

Reported CEs included constructed delirious scenarios ($n = 25$), in which the patients were trying to prevent impending danger or its

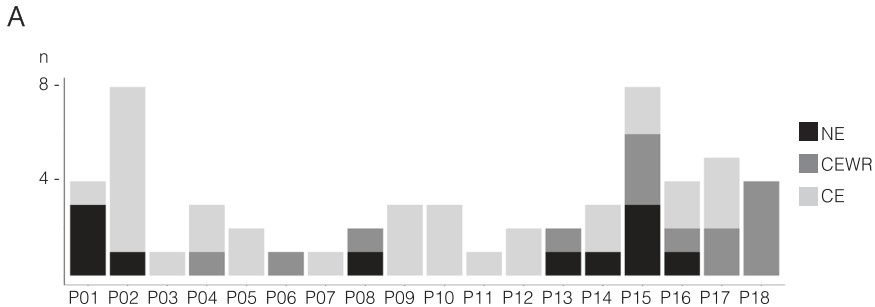

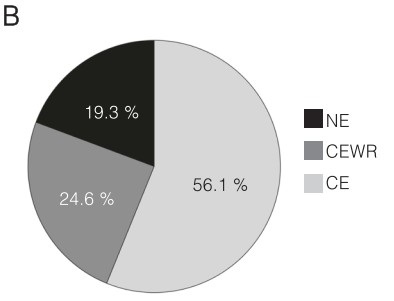

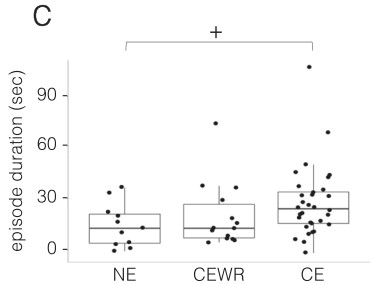

Fig. 1 | Consciousness and NREM parasomnia episodes. A Absolute number of parasomnia episodes per patient with report of conscious experience (CE), conscious experience without recall/report of content (CEWR), or no experience (NE). B Relative frequency (%) of CE, CEWR, and NE across all trials. C Duration of episodes with CE, CEWR, and NE. Each black point represents a parasomnia episode, boxes display the 25th to 75th percentile of data, horizontal bars indicate the median, and vertical bars the confidence interval. Linear mixed model: CE/NE−duration + (1|sub) and CE/CEWR−duration + (1|sub). +: $p = 0.067$ (two-sided statistics).

consequences (like needing to save ladybugs from dying, having to find one's baby daughter who had fallen off the bed, struggling because being enclosed in bedsheets, the impression that a piece of furniture was coming down from the ceiling), but also more ordinary scenarios (telling the experimenter how to fall asleep without making any sound, needing to go to an appointment with a therapist, seeing one's daughter vomit), as well as isolated imagery (a shower, green pastures, cookies, landscapes, $n = 5$) or thoughts (about taxes, bills, $n = 2$). Only two out of 18 CE reports (11%) following provoked episodes contained a possible reference to the alarm sound: in one case the patient mentioned a conversation during which she heard a sound, leading her to turn around rapidly (which she did not do in reality), and in the other case the patient mentioned "a noise, like a sound", and being scared of it, she also saw the image of an alarm. Interestingly, in some spontaneous parasomnia episodes, patients also acted as if they had perceived a sudden sensory stimulus, for instance by asking a question like: What? Hmm? or What did you say?. A clear correspondence between the reported experience and the behavior was evident in 47% of cases ($n = 15$, coherent CE), in 13% ($n = 4$) it was only partially apparent, in 34% ($n = 11$) a correspondence was not apparent, and in 6% ($n = 2$) the report was totally incompatible with the behavior (the last two categories were grouped in the incoherent CE category). A table with reports of experiences and associated behaviors is provided in Table S2.

Episodes after which patients reported NE ($N = 11$) were all but one characterized by at least one manifestation of surprise (rapid onset, startle-like movement sequence, orienting behavior, and/or perplexity), whether a sound had been played or not, and were sometimes accompanied by somniloquy ($N = 9$), a fearful facial expression ($N = 4$) and/or repetitive hand movements ($N = 2$). When asked, most patients said that they were not aware of having displayed any behavior at all ($n = 5$), while one patient reported simply having turned in bed,

although he had sat up in bed with a frightened facial expression. The category of the report (CE vs NE or CE vs CEWR) did not vary as a function of the provoked vs spontaneous nature of the episode or its occurrence during the first vs second recording, the time since lights off or the time since the first sleep onset (Table S1). Median durations of CE episodes were longer than CEWR and NE episodes, (see Table S3 but these contrasts did not reach statistical significance (CE vs CEWR: $p = 0.546$, $z = 0.603$, and CE vs NE: $p = 0.067$, $z = 1.832$, Fig. 1C). Episodes for which a clear correspondence between the CE report and the behavior was observed were significantly longer than episodes with NE ($18.8 \pm 12.15$ s vs $37.13 \pm 22.17$ s, duration: $X^2(1) = 2.24$, $p = 0.024$).

### EEG correlates of parasomnia experiences
Compared to parasomnia episodes with NE, those with CE were predicted by lower delta (1–4 Hz) and higher beta power (26–34 Hz) in posterior cortical regions in the 20 s preceding movement onset (Fig. 2A, B, left, see Fig. S1 for results at the scalp level). These differences were centered on primary visual cortices, extended inferiorly to occipital-temporal areas, anteriorly to medial temporal regions, and superiorly to include parts of the precuneus and the posterior cingulate cortex, similar to the EEG correlates previously reported for NREM and REM sleep dreams using the same methodology[34] (see overlap maps with respect to previous study in Supplementary Fig. S2 and Table S4 for cluster statistics). Differences in beta power additionally included lateral temporal areas and parts of the precentral and postcentral gyri. Parasomnia episodes with CE were also predicted, compared to those with NE, by lower delta power in posterior cortical regions after movement onset (+4 s to +20 s after movement onset, Fig. 2A right, bottom panel, see "Methods" section for the rationale behind choice of timeframes); however, compared to the previous contrast (before movement onset) these differences were mainly left-sided and comprised mostly lateral parietal and temporal areas

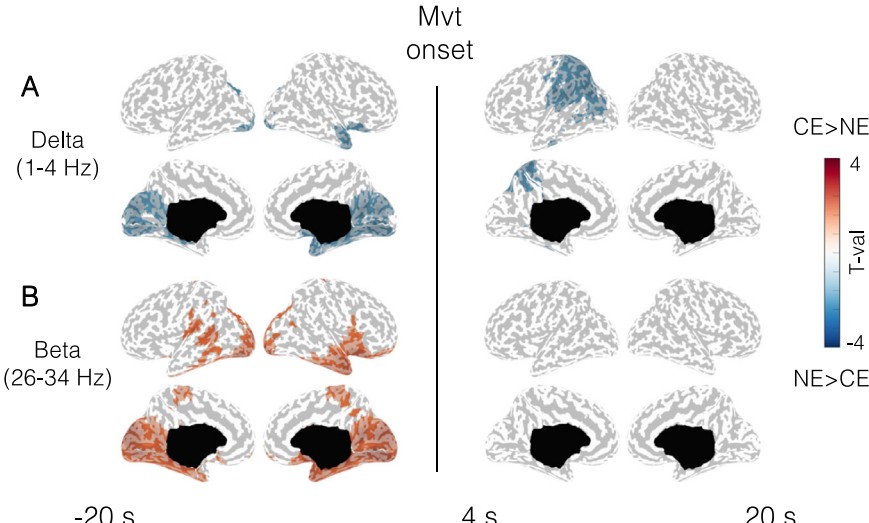

**Fig. 2 | Conscious experience (CE) vs no experience (NE): spectral power.**
**A** Cortical distribution of statistical differences in delta (**A**) and beta (**B**) power (*t* values, Wald statistics) for CE vs NE. Left column: 20 s of sleep preceding movement onset (CE: *n* = 32 episodes from 14 participants, NE: *n* = 11 episodes from 7 participants). Right column: from 4 to 20 s after movement onset (CE: *n* = 31 episodes from 14 participants, NE: *n* = 10 episodes from 7 participants). Only voxels with significant effects appear colored. LL left lateral, RL right lateral, LM left medial, RM right medial view.

(angular and supramarginal gyrus, inferior parietal lobule, paracentral lobule, and precuneus), as well as parts of the postcentral gyrus on the right side. Very similar findings to the CE–NE contrast were obtained when comparing instances in which patients reported experiences, but did not remember the content (CEWR), with NE, although results were only present prior to movement onset and were statistically weaker because fewer episodes were available for comparison (Fig. S3). Thus, decreased delta power in parieto–occipital cortices precedes behavioral parasomnia episodes with CE compared to NE, irrespective of whether the content is later remembered.

Compared to parasomnia episodes with NE, those with CE were also predicted by anterior and central slow waves with a higher amplitude, a steeper slope, and fewer negative peaks, and by fewer slow waves in posterior brain regions (Fig. S4), a slow wave constellation that also precedes reports of dreaming[12]. Similar to previous results for delta power in the present study, CE was also associated with smaller, shallower, and shorter slow waves in left posterior regions after movement onset compared to NE.

### EEG correlates of parasomnia experiences that were coherent and incoherent with behavior

We then asked whether the correlates of parasomnia experiences that were coherent with the observed behavior were different from those for which no coherence was observed. Compared to NE, both coherent and incoherent episodes displayed differences in posterior cortical regions in the scalp EEG before movement onset that were also seen when contrasting all experiences with NE, consisting in either increased beta power or reduced slow wave amplitude/slope in posterior regions (Fig. S5), although these differences were statistically weaker because fewer trials were considered, and no clear results emerged from source reconstruction maps. However, only behavioral episodes that were coherent with the reported experiences showed higher amplitude (type I) slow waves in frontal-central regions immediately prior to movement onset and lower delta power and slow wave amplitudes/slopes after movement onset compared to NE. Thus, the differences in delta power between CE and NE during parasomnia episodes (Fig. 2) were likely driven by those episodes for which the behavior was coherent with the report, while the ones for which no

coherence was observed were associated with differences well before movement onset.

### EEG correlates of recall of parasomnia experiences

Next, we tried to determine the correlates of the recall of the experience by comparing instances of CEWR and CE. Compared to CEWR, CE with recall were associated, before movement onset, by lower delta power and higher beta power in a circumscribed anterior region of the right medial temporal lobe, estimated to comprise the hippocampus, parahippocampal gyrus, and amygdala (Fig. 3A, B left). CE with recall was also predicted by lower delta power in a relatively widespread frontoparietal area after movement onset compared to CEWR (Fig. 3A right, see Fig. S6 for results at the scalp level), comprising medially the precuneus, posterior and central cingulate gyrus, the paracentral lobule and the superior frontal gyrus, extending laterally on the right hemisphere to the precentral and postcentral gyri, middle frontal gyri and the insula. The analysis of slow wave parameters confirmed that recall of the content of the experience was mostly predicted by widespread slow wave differences after movement onset during the episode (Fig. S7), while no such differences were seen in the sleep EEG.

## Discussion

Our study revealed that the degree of consciousness and amnesia associated with NREM parasomnia episodes is variable, both between and within individuals. Patients' reports ranged from NE to dreamlike scenarios characterized by delusional thinking, multisensory hallucinations, and volitional interactions with the environment. They reported a CE after 81% of episodes, could clearly recall the content of the experience in 56%, and reported NE (unconsciousness) in 19%. These results are remarkably consistent with those of a previous study documenting immediate recall of experience with clear content in 58% of NREM parasomnia episodes[12], and more broadly with the rate of dreaming reported after awakening from NREM sleep stages 2/3 when using the same methodology[40].

Most parasomnia episodes started abruptly, and were characterized, at their onset, by facial expressions of surprise and perplexity, as well as exploratory behaviors (eye opening, orienting eye and head movements). Intriguingly, these behaviors at onset occurred not only

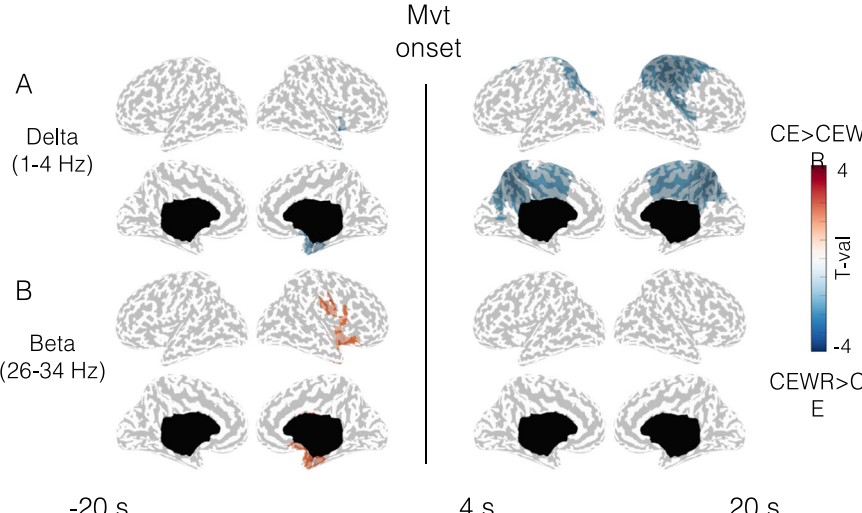

**Fig. 3 | Conscious experience (CE) vs no experience without recall of content (CEWR): spectral power. A** Cortical distribution of statistical differences in delta (**A**) and beta (**B**) power (*t* values, Wald statistics) for CE vs CEWR. Left column: 20 s of sleep preceding movement onset (CE: *n* = 32 episodes from 14 participants, CEWR: *n* = 17 episodes from 8 participants). Right column: from 4 to 20 s after movement onset (CE: *n* = 31 episodes from 14 participants, CEWR: *n* = 17 episodes from 8 participants). Only voxels with significant effects appear colored. LL left lateral, RL right lateral, LM left medial, RM right medial view.

in response to loud sounds, but also spontaneously, and perhaps even more surprisingly, sometimes without reports of experience (consciousness). More complex and idiosyncratic behaviors emerged during longer episodes and preferentially correlated with reports of experience. Thus, the initial behavior seen in NREM parasomnia episodes, which is similar across subjects (as also reported in ref. 8), likely reflects a common stereotyped activation of arousal systems, which can either occur spontaneously, or as a response to sudden arousing stimuli, and be dissociated from consciousness. The fact that almost identical behaviors, EEG patterns[36], and mental contents are observed between spontaneous and provoked parasomnia episodes suggests that provoked episodes mimic a naturally occurring arousal process[13]. Indeed, recurrent activations of arousal systems are now known to be an integral part of NREM sleep[41,42], as recently confirmed by studies demonstrating periodic activations of the locus coeruleus in slow-wave sleep[43,44]. It is therefore not too far-fetched to assume that spontaneous, naturally occurring activations of arousal systems, albeit with pathological timing[36], intensity, and/or motor coupling, could underlie the occurrence of NREM parasomnia episodes.

At the EEG level, reports of experience, compared to those of NE, were associated with activation of posterior cortical regions prior to movement onset, similar to brain activity patterns that were found to distinguish dreaming from NE in both REM and NREM sleep using the same methodology[34]. Parasomnia experiences were also preceded by large and steep slow waves in frontal and central regions, similar to the slow wave constellation that precedes reports of NREM dreaming[35] (see text S2 for further discussion).

Compared to parasomnia episodes without a report of experience, those with the report of experience not only displayed reduced delta power in the posterior hot zone before movement onset, but also after movement onset, especially in cases in which the behavior matched the report of experience. Interestingly, while before the episode differences between CE and NE were mainly localized to primary and secondary visual areas including the ventral visual stream (the what pathway) during the episode they involved more parietal, higher-order, and multisensory associative areas, and in particular the dorsal visual stream involved in spatial vision (the where pathway)[45]. It is not clear what underlies these differences in

topography, but contrary to sleep, during parasomnia episodes, by definition, patients moved around and interacted with their environment. One tentative explanation is therefore that the more consistent involvement of multisensory areas and the dorsal visual stream during the episode reflects processes associated with spatial exploration and action.

Finally, our results indicate that amnesia for the content of the experience is associated with relative right hippocampal deactivation, which may prevent the encoding of CEs (dreams) into episodic memory, as well as with persistence, after movement onset, of slow wave activity in frontoparietal regions contributing to working memory. These latter regions are again similar to the ones we previously identified as being involved in the recall of NREM dreams[34].

Taken together, our results suggest that NREM parasomnia experiences not only display the core features of dreams[25], but that they are also associated with similar brain activity patterns, which are therefore likely to reflect fundamental neurophysiological mechanisms involved in sleep consciousness[46]. Our results also raise the possibility that arousal systems could contribute to dream content. In this study we were able to induce parasomnia episodes with loud arousing sounds and observed that reported dream contents, as also described previously[8], thematically often related to impending danger or threat, suggesting a relation between the alerting nature of the stimulus and the dream content. Interestingly, the actual alarm sound rarely figured in the reports of experience, but the notion of threat did so consistently, raising the possibility that activations of non-specific (extra-lemniscal) systems, encoding the importance of the stimulus, rather than modality-specific (lemniscal) sensory pathways[47,48], relaying the precise nature of the stimulus, preferentially contribute to the CE during sleep. It is tempting to assume that a sudden activation of arousal systems, whether induced or occurring spontaneously, is interpreted as a danger signal by patients and secondarily contextualized, provided that the brain is in 'dream' or 'conscious' mode prior to the arousal and can thus 'make something' of it. Whether such a secondary enrichment of non-specific spontaneous arousal signals also underlies (NREM) dream contents remains to be elucidated.

A few study limitations deserve to be mentioned. Although this work provides one of the largest quantitative assessments of brain

activity associated with NREM parasomnia episodes, for some subgroup analyses (i.e., those on coherent/incoherent experiences), the relatively small sample size does not allow us to draw firm conclusions. In addition, parasomnia episodes mainly consisted in confusional arousals and were recorded exclusively in adults; they are therefore not representative of the full repertoire of NREM parasomnias. Parasomnia episodes also had a variable duration, but we analyzed a fixed length, which may have resulted in less clear-cut findings between categories of consciousness. Although we used the same methodology as in our previous study on dreaming, to limit the effect of movement artifacts, we did not analyze the same frequency bands and excluded the first seconds of the episode, which may have resulted in some null findings, for instance in the beta frequency band (Fig. 2). The absence of beta power differences between the CE and NE condition during the episode (as opposed to in prior sleep) may also be related to the fact that during the episode patients were moving. Indeed, beta power reduction is commonly observed during movements[49,50]. Finally, most of our results are based on associations, not allowing for causal inference. Although concordant with those of one of the largest studies on the EEG correlates of dreaming employing the same methodology, they need to be confirmed by other studies of the same type.

## Methods

### Patients

Patients presenting with disorders of arousal (confusional arousals, sleepwalking, and/or sleep terrors), as defined in the international classification of sleep disorders[1], were recruited during clinical consultations at our sleep center, or by word of mouth. All patients underwent a medical evaluation by a board-certified sleep medicine physician and a clinical polysomnography (PSG, see Table S5). One patient experienced technical issues during the PSG and underwent ambulatory polygraphy with respiratory and recording of leg movements instead. Patients who had presented at least one parasomnia episode during the last month were included. Exclusion criteria were major psychiatric or neurological comorbidities (identified during the medical consultation), medication (except birth control), and pregnancy. Participants gave written informed consent to participate in the study and to have their videos shown for educational and teaching purposes. They received financial compensation for participation. The study protocol was approved by the local ethical committee (commission cantonale d'éthique de la recherche sur l'être humain du canton de Vaud) and was carried out in compliance with all relevant ethical regulations.

### Experimental procedure

Patients underwent a first (baseline) hd-EEG nighttime recording between ~11.30 p.m. and ~6.30 a.m., during which they were continuously monitored via video and audio by an experimenter. The next day they were free to perform their usual activities but were not allowed to sleep. Compliance with the instructions was verified by wrist actigraphy. In the evening, they came back for a night of supervised total sleep deprivation in the laboratory. A sleep technician verified, via continuous video and audio recordings, that they did not fall asleep. The following morning, a second hd-EEG (recovery) sleep recording was carried out between ~7.30 a.m. and ~4 p.m. During this second recording, computerized pure sounds (1000 Hz, lasting 3 s) were played at one-minute intervals after five minutes of stable slow wave sleep, with increasing intensities (50–90 dB, increase of 5 dB between each interval, adapted from[38]), until they provoked a parasomnia episode or a full awakening. This procedure has previously been shown to increase the frequency and complexity of NREM parasomnia episodes. During both the baseline and recovery sleep recordings, when a parasomnia episode occurred, the experimenter waited for the episode to end, called the patient's name, and then conducted a semi-structured interview (based on refs. 34,40) through

intercom about the patient's subjective experiences. If the patient did not react immediately, the experimenter called her/his name a second time. Patients were asked what they had experienced immediately before the examiner had called their names. When they did not report a CE, they had to clarify whether they had experienced something but could not recall/report the content of the experience (CEWR), or whether they had not experienced anything (NE).

### Rating of parasomnia episodes

The procedure for defining and rating of parasomnia episodes has been described previously[36]. More specifically, the video and EEG of all awakenings out of slow-wave sleep were extracted, regardless of whether they were associated with manifestations of parasomnia. The beginning of an episode/awakening was defined as movement onset on EMG or video, whichever occurred earlier, while the end was defined as movement cessation. In most cases, at the end of an episode patients paused, then sometimes started an interaction with the examiner to signal that they were fully awake, or resumed a sleeping position again. Three experts independently rated the behavior seen on the videos (without the EEG) as either a parasomnia episode (as defined in the ICSD[1]) or a normal awakening. Only episodes considered as such unanimously by all the three raters were included in the analyses. Experts also rated the presence of the following features: interactions with the environment, somniloquia, manifestations of fear and of surprise (rapid onset, startle-like movement sequence, orienting behavior, and perplexity), and, when a mental content was reported, whether it was coherent with the behavior seen on the video or not. A feature was considered present when at least two of the three judges had rated it as such. The coherence between the report and behavior was rated separately by the judges. In case of non-agreement, a discussion between the raters followed until agreement was reached. This method was chosen to reduce the complexity of the scoring procedure and to separate coherent and incoherent behavior without further reducing the number of trials.

### EEG recordings and analysis

**EEG recordings.** Sleep was recorded with a hd-EEG system (256 channels, Electrical Geodesics, Inc., Eugene, Oregon) and a 500 Hz sampling rate, in a certified sleep center, guaranteeing conditions that are appropriate for sleep recordings. The EEG cap was connected to a 4-m-long cable that allowed patients to leave the bed and move around the room. Four electrodes located near the eyes were used to monitor eye movements, and electrodes overlying the masseter muscles to record muscle tone[13].

**Preprocessing.** EEG recordings underwent a thorough artifact removal procedure (Fig. S8). First, the EEG was bandpass filtered between 0.5 Hz and 35 Hz. Then, the EEG corresponding to parasomnia episodes was extracted, as well as the EEG of the five minutes of sleep preceding movement onset. To remove additional artifacts, artifact subspace reconstruction (ASR)[51] was applied to the EEG corresponding to the parasomnia episode after movement onset. ASR is an advantageous method due to its capability to efficiently and automatically eliminate non-stationary artifacts of significant magnitude, irrespective of their distribution on the scalp or their consistency across the dataset[51–55], and was recently validated for sleep[56]. It works as an adaptive spatial filtering algorithm, which uses clean sections of EEG signal to produce a statistical model to reject artifactual EEG segments based on how much they deviate from the calibration signal in the principal component analysis subspace[57]. The artifact-free sections were automatically derived by the algorithm, based on standard deviations from a robust estimate of the EEG power distribution in each channel. A moving window of 768 ms (1.5× the number of channels) and a 25 standard deviation cutoff for rejection were used to avoid rejection of sleep features (such as slow waves persisting during

the awakening/episode). The threshold was determined based on visual inspection to maximally remove noise and completely preserve sleep slow waves[51]. EEG analyses at the scalp level were performed on the innermost 175 channels to further limit artifact contamination. Artifactual channels were visually inspected, marked, and interpolated with data from other channels using spherical splines (NetStation, Electrical Geodesic Inc). To exclude ECG and other artifacts, including ocular, movement, and electrodermal activity, independent component analysis was performed using EEGLAB routines on sleep and parasomnia episodes separately[57,58].

**Spectral power decomposition.** The spectral power decomposition of the EEG signal was computed on a 40 s timeframe centered on movement onset[13], for non-overlapping 2 s and 0.5 s epochs in the delta (1–4 Hz) and beta (26–34 Hz) frequency bands, respectively. The frequency bands were chosen as representative for low-frequency power (slow wave activity) and high-frequency power, respectively, based on our previous study on dreaming[34], and to simplify the presentation of results. Analyses of spectral power were based on the 20 s of sleep preceding movement onset in light of our previous studies on the EEG correlates of dreaming[13], and on the 20 s window after movement onset, for symmetry and because this timeframe approximately corresponded to the median duration of parasomnia episodes. The first four seconds after movement onset were excluded from analysis because of the frequent presence of residual movement artifacts in the EEG. Thus, the analysis for the EEG after movement onset was restricted to the timeframe ranging from +4 s to +20 s after movement onset.

**Source modeling.** Source localization was performed using the GeoSource software (Electrical Geodesics, Inc., Eugene, Oregon), on filtered and preprocessed EEG segments. Individualized geocoordinates of the electrodes were used to construct the forward model. The source space was restricted to 2447 dipoles distributed over $7 \times 7 \times 7$-mm cortical voxels. The inverse matrix was computed using the standardized low resolution brain electromagnetic tomography constraint[59]. A Tikhonov regularization ($\lambda = 10^{-2}$) procedure was applied to account for the variability in the signal-to-noise ratio[59].

**Slow wave analysis.** A previously validated slow wave detection algorithm[12] was applied to the EEG signal. Specifically, the EEG signal was baseline corrected, subtracting to each electrode signal from its mean. Then it was re-referenced to the average of the two mastoid electrodes, downsampled to 128 Hz and bandpass filtered (0.5–4 Hz; stop-band at 0.1 and 10 Hz) using a Chebyshev Type II filter (Matlab, Mathworks version 2015a). The slow wave detection was performed for each channel separately and consisted in identifying the zero crossings. The negative deflections between two zero crossings were defined as half-waves. Only half-waves with a duration between 0.25 s and 1 s were considered, and no amplitude threshold was applied. After applying a moving average filter of 50 ms, the determination of negative and positive peaks was based on the zero crossings of the derivative of the signal. Then, for each slow wave, different parameters were extracted: maximum negative peak amplitude, slope 1 (between the first zero crossing and the negative peak), slope 2 (between the negative peak and the second zero crossing), the number of negative peaks and the duration (time from the first and second crossing). Finally, the parameters of each slow wave were averaged and the density (expressed in the number of slow waves per minute) was extracted.

**Statistical analyses.** Statistical analyses were performed in R-Studio (version 1.1.463). Separate generalized linear mixed models were used to predict the presence of CE vs NE or CEWR as a function of different variables, including the provoked or spontaneous nature of the episodes, the occurrence during the baseline or recovery sleep, the time passed since lights off, and the duration of the episodes. A

generalized linear mixed model was also used to predict the spontaneous vs provoked nature of the episode as a function of manifestations of surprise. For this model, a surprise index, ranging from 0 to 4 was calculated for each episode, with each of the following manifestations contributing one point to the score when it was present: sudden onset, startle-like movement sequence at onset, orienting behavior, perplexity. Subject identity was always included as a random factor in all models.

To investigate how EEG activity predicted different categories of reports (CE, CEWR, NE, etc.), generalized linear mixed models were computed for each electrode (scalp) or voxel (source) and timeframe (before vs after movement onset). The frequency bands (delta and beta) were included as fixed factors, and subject identity and the nature of the awakening (spontaneous/provoked) as random factors. The Wald statistics (the squared ratio of the fixed factor estimate over its standard error) were obtained for each model. For these analyses opted for the one-sided statistics because we had a priori hypothesis, based on our previous work about the direction of results. In this sense, the p-values resulting from the model were therefore divided by two. A cluster and probability-based correction was applied as follows: a dummy population was created by shuffling the labels between report categories 1000 times. For each permutation, the model was applied and neighboring electrodes with p values < 0.025 or 0.05 (according to one or two-tailed statistics) were identified as a cluster. Electrodes/voxels clusters were defined spatially, and neighbors were defined through a function from the Matlab fieldtrip toolbox using the triangulation method, which calculates a triangulation based on a two-dimensional projection of the sensor position. A cluster statistic was produced by averaging Wald statistics within each significant cluster and the maximal absolute cluster statistics were retained for each permutation. The real cluster statistics obtained in the real dataset were compared with the threshold of significance, which was set at the 97.5th or 95th percentile (according to one- or two-tailed statistics) of the dummy cluster statistics distribution. The generalized linear mixed model formulas and the voxel/channels statistics are indicated in Tables S1 and S3, respectively.

**Reporting summary**
Further information on research design is available in the Nature Portfolio Reporting Summary linked to this article.

## Data availability
The data generated in this study are provided in the Source data file. Videos of parasomnia episodes have not been included in this publication to protect patient privacy. However, they can be viewed upon direct request to the corresponding author, as the patients have given written informed consent for their videos to be viewed for educational and scientific purposes. Source data are provided with this paper.

## Code availability
Templates of the custom-made codes in Matlab, Mathworks (version 2015a), and R-Studio (version 1.1.463) are available at 'https://github.com/jcataldi2/SleepWalking/.

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

## Acknowledgements

This work was supported by grants from the Swiss National Science Foundation (Ambizione Grant PZ00P3_173955), the Théodore-Ott Foundation, and the Foundation for the Advancement of Neurology, awarded to F.S. The authors thank the patients for participation in the study, and, in alphabetical order, David Albir, Françoise Cornette, Stephanie Dutoit, Raphaël Heinzer, Eric Lainey, Gianpaolo Lecciso, Tifenn Raffray, Geoffroy Solhelac, and the entire CIRS team for patient referral and help with data acquisition, as well as Jean-Baptiste Maranci for his comments on the manuscript.

## Author contributions

Conceptualization, F.S.; methodology, F.S., J.C., and J.H.R.; software and formal analysis, J.C. and A.M.S.; investigation, F.S. and J.C.; writing (original draft), F.S. and J.C.; writing (review and editing), all authors; visualization, J.C.; supervision and project administration, F.S.; and funding acquisition, F.S.

## Competing interests

The authors declare no competing interests.
