## [Peer Review File · Nature Communications]

Shared EEG correlates between non-REM parasomnia experiences and dreamsREVIEWER COMMENTS

Reviewer #1 (Remarks to the Author):

In this study, Cataldi and coll. investigated the neural signatures associated with sleepwalkers' mental experiences (while asleep) and with the recall of such experiences. The study builds on previous findings from the same group, where they found that a local activation (decrease in low-frequency activity, increase in high-frequency activity) in a 'posterior hot zone' during sleep was associated with subsequent reports of mental experiences (a.k.a. dreaming). Here, they found that patients with sleepwalking display a similar activation of the posterior hot zone (before and during the episodes), even when they did not remember the dream. Amnesia of the dream content was associated with fronto-parietal slow wave activity during the parasomniac episode.

I really enjoy reading this manuscript, which is very clear, well-written, and tackles a topic that has been rarely investigated. The study is cleverly designed and the analyses are appropriate, with top-of-the-art methods. Given the difficulty to obtain sleepwalking episodes in the lab, the amount of data collected here is truly impressive. By recording high-density EEG (instead of a few electrodes typically used in clinical settings) and collecting dream reports right after the episodes, the authors have acquired an original, one-of-a-kind dataset. This tour de force is likely to have a massive impact, both on basic researchers interested in consciousness/dreams and on clinicians.

Major comment:

While I found the study fascinating, I was left a bit confused on how the objectives were framed and felt that I missed a brick in the reasoning. Basically, I'm unsure whether the main goal was:

1) To empirically confirm that sleepwalking episodes can be viewed as dream enactment.

This question remains controversial and some authors believe that sleepwalking episodes are automatic and do not reflect dreams (e.g., Tassinari et al., 2009)

2) Test the 'chicken or egg problem': does the dream cause the arousal or the arousal the dream?

I felt that the manuscript subtly oscillated between these two goals and that the analyses

and/or results interpretation could change depending on which one is chosen.

In the case of objective 1 (hyp: sleepwalking = dream-enactment), I feel some discussion points are missing. Indeed, in their Nature Neuro study, the authors computed the neural signatures of dreaming right before the dream report, so supposedly at the very moment when the conscious experience occurred. But here, if one wants to prove that sleepwalkers do dream during the episodes, the main period of interest is the parasomnia episode itself (which, as the authors elegantly say, 'timestamps' the experience). However, the EEG analyses computed during the episodes show less robust results than the ones computed before. For example:

- The contrast CE vs. NE was associated with lower delta power in posterior cortical regions during the episode, but not exactly in the same regions. This is cool and reassuring. But contrarily to the previous study, there were no differences in beta activity between CE and NE. Why?

- The contrast CE vs. CEWR during the episode does not show the same results as the contrast CE vs. NE contrast => in that case, it is difficult to be sure that the conscious experiences reported by patients truly correspond to the episode and not to the dream that happened before (but I agree that we can be more confident when the external behavior matches the report)

- The amplitude of delta waves (+ Slope I and slope II) seems to show an opposite pattern before and during the episode when comparing CE vs. NE (Fig 3 and 4). If higher amplitude (type I) slow waves in fronto-central regions reflect arousal before movement onset, does it mean that patients who report CE are less aroused during the episode than the ones who do not report any experience (NE)? Please discuss.

- In general, if the goal was to prove that sleepwalkers are dreaming during the episodes, I am unsure how to interpret the pre-episode EEG signatures of dreaming. Are they showing that the parasomnia episode is simply continuing the dream occurring just before the awakening? Or is the episode a new dream?

In the case of objective 2 (chicken and egg problem), there are two hypotheses (dream causes arousal or arousal causes dreams). It is unclear how the three specific aims stated at the end of the intro (lines 119-123 p4) will help validate or reject these two hypotheses. Throughout the manuscript, it seems that the main strategy to test the chicken and egg problem is to compare spontaneous vs. provoked episodes. However, it is unclear how exactly this comparison will allow to solve the problem. For example, lines 243-244 is the first mention of an analysis predicting the spontaneous v. provoked nature of the episode as a function of surprise manifestations. But it is unclear until the results section what was the background/ hypothesis behind this analysis.

What I'm missing is clear predictions spelled out at the end of the intro, such as: if dream causes arousal, we would expect A and B. By contrast, if arousal causes dream, we would expect C and D. On an unrelated note, why would arousal and dreams necessarily be related? Could it be that sleepwalkers are dreaming like anybody else and that, at some point, an arousal releases the inhibition of motor outputs, allowing sleepwalkers to enact their ongoing dreams in a similar manner as in patients with RBD?

Finally, if the main question is to investigate the link between arousal, parasomniac episode, and dreaming, additional analyses could be considered. For example:

- What happens in the dream hot zone when sounds do not trigger an episode?
- What happens in the dream hot zone before normal awakening from N3 (i.e., the 25% of the episodes that were not rated as 'parasomniac episode'?)
- Why would activation of the hot zone (arousal) provoke the episode? Do you expect such arousal to be more intense than in healthy subjects? If so, could you compare the data in sleepwalkers with the ones from your previous study in healthy subjects?

To make a long story short and condensate this long comment (sorry!), could you clarify what the objectives and hypotheses were and discuss the results according to the initial expectations?

Minor comment:

- To ensure the replicability of the results, please detail the methodology further. In particular:

- The recovery recording was set between 7.30 and 4pm => it is not a typical time for sleepwalking episodes. Was this time chosen for practical reasons or is there a scientific reason?

- L155 p5: How was the 'supervised total sleep deprivation' conducted?

- What happens if participants did not wake up once the experimenter calls their name?

- L176: end of sentence missing => episode rated as 'either a parasomnia episode' or? Please detail the criteria used to define the episodes.

- L 177: judgment on specific features associated with the episodes (e.g., interaction with the environment, somniloquia, etc) => please detail the criteria. Were there several, independent judges? If yes, what is the inter-scorer agreement?

- Similarly, how did you judge the coherence between dream report and parasomniac episode? Several methodologies have been used in the literature (e.g., mixing up the dream reports and have a few judges choose the one that matches the behavior the most; have several judges indicate a score of coherence on a scale between 0 to 10, etc). Each method comes with its own biases (no ideal method). Please justify why you choose one approach rather than another.

- L188: why use the masseter muscle and not the chin as usually done in sleep studies?

- L 199: since obtaining clean EEG data in moving participants is a real challenge, the methodology developed here could be very useful for researchers from different fields. Could you elaborate more on the procedure for rejecting movement artifact and provide a figure illustrating the procedure as well as snapshots before/after?

- When the patients reported no mental experience, did they remember having moved?

- Other comments:

- The number of episodes seems to be the same during the baseline (n=37) and recovery night (n=38). Does it mean that Pilon's procedure to trigger episodes is actually not working? One would have expected a lot more episodes during recovery night than at baseline. Given the cost (money- and time-wise) of conducting a full sleep deprivation protocol in sleepwalkers, it might be important to inform the sleep community of the limits of such procedure.

- Fig 2A: I see no significant cluster between CE and NE during parasomnia episode (delta power), but a significant difference is mentioned in the main text (l358-361).

- L 360: there is a typo => Fig 2A left should be Fig 2A right

- Please put the corresponding legend below the figure to make readers' job easier and avoid back and forth.

Reviewer #2 (Remarks to the Author):

In this very interesting study, Cataldi and colleagues: 1) systematically collected parasomnia experiences immediately after spontaneous or provoked NREM-related parasomnia episodes and; 2) compared EEG brain activity immediately before and during episodes with and without conscious experiences. Conscious experiences were reported in 83% (and a specific content in 56%) of the 57 recorded episodes followed by a semi-structured interview (collected from 18 subjects). Compared to reports of "no experience", reports of "experience" were preceded by an activated EEG pattern over posterior cortical regions (lower delta and/or higher beta power). A similar pattern persisted during the episodes when patients displayed complex behaviors and conscious experiences. In addition, content amnesia appeared to be modulated by the degree of right hippocampal activation during prior sleep and the persistence of fronto-parietal slow wave activity during the episodes. These results were interpreted in the light of a previous paper on the neuronal correlates of dreaming, published by the same group. The authors concluded that their findings suggest that parasomnia experiences share the same neural correlates as dreams.

Overall, the manuscript is well written, the aims of the manuscript are clear and logically

presented, the list of references is up-to date, the discussion is appropriate, and the statistical analyses appear methodologically sound. However, I have several remarks the authors may want to address before the manuscript could be considered for publication.

- Abstract:

- I recommend to add a statement on the frequency of conscious experiences reported immediately after parasomnia episodes, as this was listed among the main aims in the introduction, and it is actually a very interesting data.

- I also recommend to report the specific brain areas found to be significant in the comparison between “experience” vs “no experience” episodes and to talk about the “dreaming signature” interpretation afterwards, in order to make a clear distinction between results and the conclusions driven by the authors.

- Line 42, page 2: The number of total “potential” episodes was 102, but the number of “sure” episodes was 75, and the number of “sure” episodes followed by an adequate interview was 57. Thus, reporting just the number of 102 may be misleading.

- Line 49-50, page 2: It is not entirely clear to me how the sentence “and that arousal systems may play a role in the generation of dream contents.” could be justified by the results summarized in the abstract (where slow waves type I and II are not mentioned).

- Introduction:

- It may be worth mentioning among the aims, or as a secondary aim, the comparison between spontaneous and provoked episodes as a function of manifestations of surprise.

- Furthermore, there is no mention to the slow wave detection analysis, its background and rationale (neither here nor in the discussion but only in the result section).

- Methods:

- If the authors have used the same (or overlapping) dataset of their recently published paper in Sleep, they would need to state this point explicitly here.

- Line 144-146, page 5: How did the authors exclude major psychiatric or neurological comorbidities, sleep apneas with and AHI >15/h, and/or a periodic leg movement index in sleep >15/h?

- I suggest to add, at least in the Supplementary Material, a table summarizing common

sleep variables, such as TST, TIB, % and min in each stage, AI, PLMI, AHI.

- Line 175, page 6: “while the end was defined as movement cessation”: what if an episode ended in wakefulness, as probably always happened for the analyzed episodes? This is a tricky point. Please provide more details.

- Line 195, page 6: regarding artifact subspace reconstruction (ASR) > was this method (and its related parameters) previously validated for sleep? Did you visually inspect each segment after this automatic procedure? May you please provide some examples of the EEG signal before and after the procedure in the Supplementary Material? This may increase the trustworthiness of your procedure.

- Line 205, page 7: EEG analyses at the scalp level were performed on the innermost 175 channels to further limit artifact contamination > I may agree, but please provide the readers a rational why this was done only for scalp and not for source analysis.

- Line 206-207, page 7: “Artifactual channels were visually inspected, marked and interpolated with data from other channels using spherical splines (NetStation, Electrical Geodesic Inc): was this done after ASR? If so, how did you ensure that the reconstructed signal was not reconstructed using artifactual channels?

- Line 208, page 7: To exclude ECG and other artifacts, independent component analysis (ICA) was performed > Which other artifacts? Please specify. ICA is a powerful tool, but removing apparently “bad” components (without a clear artifactual meaning) looking only at their topography, may lead to the removal of real EEG signal.

- Line 211-216: were other frequencies computed? If yes, they should be presented in the Supplementary Material. Please, consider also explaining why you selected delta and beta in order to make the rational clear to the readers - who may not always be expert in the topic).

- Line 255-256, page 8: The Wald statistics (the squared ratio of the fixed factor estimate over its standard error) were obtained for each model, both with an alpha level of 0.05 and 0.1: I could not see why a threshold 0.1 should be considered here. Please provide a rational, or (better), just avoid mentioning the 0.1 threshold here in Methods and in the Figures/Results.

- Line 345-348, page 11: “Analyses were based on the 20s of sleep preceding movement onset (“sleep EEG”, in light of our previous studies on the EEG correlates of dreaming), and on the 20s window after movement onset (“parasomnia EEG”, for symmetry and because this timeframe approximately corresponded to the median duration of parasomnia

episodes)” > I have one major concern here. You stated that “Episodes for which a clear correspondence between the report and the behaviour was observed were significantly longer than NE (18.8 +/- 12.15 secs vs 37.13 +/- 22.17)”. This might represent a bias when comparing these two conditions. Please either list this issue among the limitations, or to add an analysis on shorter and more comparable segments.

- Line 345-348, page 11: “The first four seconds after movement onset were excluded from analysis because of the frequent presence of residual movement artifacts in the EEG.”: are these 4 seconds part of the 20 second window after motor onset (meaning that the analyzed segments were of 16 seconds after all)?

- Results:

- Please, if possible, provide more clinical details regarding the patient population (e.g., comorbidities, DoA familiarity, age at DoA onset, DoA subtypes).

- Line 334: There was a trend towards a longer duration for CE compared to NE episodes ($p = 0.067$, $z = 1.832$), but not for CE compared to CEWR episodes ($p = 0.546$, $z = 0.603$) (Fig. 1C).> please provide also mean/median values and standard deviations/ranges.

- There are several repetitions between methods and results regarding the description of the statistical models, I suggest to simplify the text.

- I may have missed this information: was there a difference in the time from onset between episodes with or without mental recall?

- Line 390, page 12: EEG correlates of parasomnia experiences that were coherent and incoherent with behaviour: why there are no source statistics for this specific analysis?

- Figures

- Nose and ear edges seem to be cut out of the print. I suggest to choose better plotrad, intrrad and headrad for figures 2-6 and s1-2.

- Some colorbar units in figures 2-6 are messed up (e.g., W/min). I suggest to make them slightly smaller.

- P values <0.1 have no statistical meaning. I suggest to avoid plotting these values from figures 1-6 and s1-2.

- The blue bar representing sleep at the top of many plots in Figures 2-6 and s1-2 is slightly asymmetric compared to the mvt onset and between plots.

- Language:

- I recommend the authors to go through the text accurately and make minor stylistic changes and grammatical/punctuation corrections. Below, only a few examples.

- Line 74, page 3: there is a space between the word “consciousness” and references 1-2, while for subsequent references no space is used > please revise the references according to the Journal guidelines.

- Line 89, page 3: I am not sure why you used square brackets without content [...], in the middle of this long statement.

- Line 94, page 3: For one > First.

- Line 110, page 4: In the present study: If starting the sentence with a prepositional phrase, then a comma should separate the phrase from the rest of the sentence.

- Line 122, page 4: “EEG signatures of dreaming 3) to clarify whether the” > a comma is missing here. Consider the use of semicolons instead of commas.

- Line 145, page 5: medication > medications.

- Line 175-177, page 6: Three experts independently rated the behaviour seen on the videos as either a parasomnia episode (as defined in the ICSD2). > Perhaps an “or” is missing here.

- Line 197, page 6: please use “that” in defining clauses, “which” in non-defining clauses (preceded by a comma)

- Line 229, page 7: algortihm > algorithm

- Discussion:

- Line 450-452, page 14: “a previously documented electrophysiological pattern that was found to distinguish reports of dreaming from those of no experience across REM and NREM sleep”: looking at the previous paper, I am not fully convinced that the brain regions are exactly the same, especially regarding beta power, but it is not easy to understand from the figures. Maybe you could specify overlapping and not overlapping regions (if any).

- Line 452-455, page 14: “Parasomnia experiences that were coherent with the observed behaviour not only displayed the ‘dream signature’ during prior sleep, but also during the episode compared to NE (in the form of reduced delta power in posterior cortical regions), as if the brain remained in ‘dream mode’ after the initial arousal.”: This is very interesting, however, I have two questions: 1) How do you explain the difference between the pattern observed prior to motor onset and the pattern after motor onset (clearly more anterior for

delta power?); 2) where are the statistics/figures for the source analysis?

- Line 478-479, page 15: "In conclusion, somnambulistic experiences not only share core features of dreams, but also display similar neurophysiological correlates as dreams." >

Perhaps, as you previously suggested that the 'posterior hot zone' constitute a core correlate of conscious experiences in sleep, what is mainly shared here, is the fact that patients are conscious before and during episodes. Although, yes, this does not completely fit with the fact that during episodes the posterior hot zone seems slightly more anterior (see previous point).

- If I understood correctly, in your previous paper, the EEG activity during episodes and during physiological movements in patients did not differ. Did you collect mental content after physiological movements? What would you expect? Could you please comment on this point?

- Existing controversial points (e.g., other findings about neural correlates of dreams by other groups), limitations and future research needs could be discussed more thoroughly.

Reviewer #3 (Remarks to the Author):

Review of Non-REM parasomnia experiences are associated with EEG signatures of dreaming

In this article, Cataldi et al. explore the neural correlates of NREM parasomnia events, and the content of these states (i.e. dreaming, no-dreaming). They further attempt to classify whether the behaviours during a parasomnia is associated with dream content. While an interesting topic, I have some reservations regarding the setup, the analysis and the interpretation of these results, as specified below.

Introduction:

I like the inclusion of 'pop' Shakespearean topics, but I am sure there is an actual reference to sleepwalking that would be more pertinent to a science journal.

If you are going to call sleepwalking by it's common name, keep it, or at least make reference to the fact that it means 'somnambulism.'

1960's? I don't think I have ever seen it written 1960ies.

The debate of whether or not dreaming occurs during sleeping, or is a confabulation, is interesting, but it doesn't really have to do with the article at hand. Like you say at the beginning of the second paragraph, there is no more doubt about whether dreams happen in NREM sleep. This whole debate doesn't set up anything to do with your article.

Page 3, line 97-98, maybe rephrase, difficult to parse. It seems like you're saying that activity related to some event is present during sleep, and during wake, and therefore we can say they reflect the same experience. But the whole point of neuroimaging is that it is correlational, you can't draw causation from such things.

Line 103-104; same deal; EEG hallmarks of sleep occur while people are awake, and therefore these must relate to sleep? This is again drawing causation from a correlative measure. I am not saying that these parasomnias aren't sleeping states, but placing the argument as, they are awake, but have sleep signals in the brain, so these co-occur doesn't really make sense to me. Could it be that these people are asleep? This whole thing seems rooted in a debate on the nature of 'awake' and consciousness more generally. I'd avoid the whole debate.

Methods

SWS? Please define acronym.

For most of your analysis steps, the choices seemed pretty standard. I was curious what the purpose of the slow wave analysis was, specifically, and why so many different parameters were extracted. Do you have a theory as to why each of these would be interesting, perhaps correlating with something to do with sleep? Otherwise it seems like fishing...

The slow wave analysis could be described in more detail. With the limited work I have completed with this (a few years ago) using the WISPIC toolbox, there are more arbitrary choices for the definition of a slow wave that are not sufficiently reported. It is also unclear what this adds over delta power, my impression is that it does not add much.

Stats, was it CE vs (NE or CEWR) or CE vs NE vs CRWR? Surely CEWR and CE would be more similar as a grouping than NE and CEWR.

Please report your formula for GLMMs somewhere. A simple description doesn't really cut it; for example, what are main effects, what are included as interactions etc etc.

0.1 is a high cluster inclusion alpha. Usually, I'd use 0.05 or 0.025.

It sounds like you did a cluster-based permutation test. Did you use a function for this, or calculate it all manually? How did you define clusters (temporally, spatially, both?), you mentioned electrodes, but how did you decide on neighbouring electrodes? For example, I'd normally use fieldtrip to get a list of neighbouring electrodes based on distance.

Results

How did you confirm that the unprovoked parasomnia events were unprovoked? Were the rooms sound proof etc etc.

You note that the types of episodes are behaviourally indistinguishable. Is it possible that the two types of events are both caused by some type of stimulation, however we just don't know the cause of the 'spontaneous' awakenings?

Line 334; there wasn't a trend; p-values are an (arbitrary) cutoff we use in science, things are either significant or not significant.

Did you look at the beta/delta ratio for eeg activity? Ratio of high to low frequency activity is pretty well established in, sleep (and your prior) research.

Line 370-377: this is the type of context that would have been helpful above where I asked why you looked at these various slow-wave effects.

For all these EEG stuff, where are the stats? I see in the figures you gave us p-value stats, but a) you should be giving cluster statistics as well, and b) if you are doing a cluster-based permutation analysis, you should be reporting a p-value for a given cluster, rather than

individual electrodes (your image could be confused in this way). Plus, you should report precise p-values unless $p < .0001$ or so; $p < .05$ is something out of the 80s.

Line 400- 404; you could test for this statistically. If you don't, drop the line. Causal inference from one thing not happening is really weak. Also, as an aside, because using negative inference as you have done here begs the question; what kind of power did you have for this study?

I think a major issue is whether 22 people, with 107 events to analyze, was enough to find a difference in the perceived behaviour reported by the technician analyzing said behaviour and the (obviously imperfect) report of a patient who just woke up from a parasomnia event? This is very important given the heterogeneity the authors emphasize.

In this results section you have done a lot of 'trying to predict x from y' type of analysis. Did you consider perhaps a machine learning approach, rather than GLMMs?

How much do you trust deep structure analysis like hippocampal and amygdala reconstruction from EEG? If each patient didn't have an MRI (and really, probably even if they did) using a generic head model is probably going to lead to unreliable results in this way. You can get 'ghost' deep sources due to the math of how source reconstruction works (electrical spreading etc.).

Discussion

56% as consistent with the 'rule' rather than the exception isn't really a strong rule.

Line 460- going from activation of the arousal system to an interpretation of a danger signal might be right; but if you are arguing that the dreams are online rather than post-hoc confabulations; how can we tell? Maybe we get a strong arousal signal, and then when we wake up, we have to make up a reason for why we had this arousal? Like the classic neurostimulation experiment of the girl who would laugh if her SMA was stimulated, and would, post-hoc, make up reasons for why this was.

Do you think all this analysis of EEG was warranted, given the limited scope of what you report in the discussion? I would focus on the power analyses.

477- why do you keep using the phrase 'somnambulistic;' as you report here, no one left

their bed during the experiment.

479 – so your answer to the chicken and egg question is, why not both? Maybe a different analogy is needed. That the effect is somewhat cyclical or builds up makes intuitive sense though.

This final paragraph makes much more sense and clarifies what you're trying to say.

Basically, you're reporting there is a potentiation, and then there is an arousal. If the potential is low, a large arousal would be needed to generate a parasomnia, but people might wakeup first. If the potential is high enough, even an indiscernible arousal could cause a parasomnia event. Maybe make this clearer earlier on, however this is unlikely to be unique to parasomnia and hence I think the unique insights for parasomnia are limited.

Reply to comments of referees

We are indebted to the reviewers for their thorough revision of the manuscript, their insightful remarks and stimulating questions. We have extensively revised the manuscript based on their comments and are confident that it has greatly improved as a result. Major changes are summarized below. A point-by-point reply follows. For ease of review, the modified parts of the text have been included in the responses whenever possible and appropriate and appear in italics with changes underlined.

- 1) From the reviewers' comments we gathered that it was not clear which the main aims and hypotheses of the study were, and that the *introduction* was too broad and *not focused enough*. We have therefore completely rewritten and simplified the introduction, focusing on two main aims (quantifying conscious experience after parasomnia episodes and correlating reported conscious experiences with regional brain activity), and removed context that was not strictly related to the main study aims. We have also simplified the figures, focusing on the main results that are relevant to the primary aims, and moved other results to the supplementary material.
- 2) The reviewers asked for a *more precise comparison with our previous findings*. We now provide comparative maps in the supplementary material and have discussed the differences more extensively in the discussion section.
- 3) The referees asked whether *the slow wave analyses were justified* and wondered about their added value. We have now introduced the rationale for the slow wave analyses in the introduction and discussed them more in depth in the discussion section. We have also simplified the figures, showing only the most important slow wave parameters and moving the remaining ones to the supplementary material.
- 4) Following the reviewers' suggestion, we have now included *more information regarding the artifact removal procedure* and provided a figure illustrating it in the supplementary material.
- 5) A paragraph describing the *major limitations of the study* has been added to the discussion, mentioning the limited power for some subgroup analyses and other methodological limitations that may have resulted in some null findings, etc.
- 6) The *title* has been slightly modified.

Reviewer #1 (Remarks to the Author):

- 1. In this study, Cataldi and coll. investigated the neural signatures associated with sleepwalkers' mental experiences (while asleep) and with the recall of such experiences. The study builds on previous findings from the same group, where they found that a local activation (decrease in low-frequency activity, increase in high-frequency activity) in a 'posterior hot zone' during sleep was associated with subsequent reports of mental experiences (a.k.a. dreaming). Here, they found that patients with sleepwalking display a similar activation of the posterior hot zone (before and during the episodes), even when they did not remember the dream. Amnesia of the dream content was associated with fronto-parietal slow wave activity during the parasomniac episode.**

I really enjoy reading this manuscript, which is very clear, well-written, and tackles a topic that has been rarely investigated. The study is cleverly designed and the analyses are appropriate, with top-of-the-art methods. Given the difficulty to obtain sleepwalking episodes in the lab, the amount of data collected here is truly impressive. By recording high-density EEG (instead of a few electrodes typically used in clinical settings) and collecting dream reports right after the episodes, the authors have acquired an original, one-of-a-kind dataset. This tour de force is likely to have a massive impact, both on basic researchers interested in consciousness/dreams and on clinicians.

We thank the referee for the insightful comments and stimulating questions. We are pleased to read that the large amount of work behind the acquisition of this dataset is acknowledged.

Major comment:

- 2. While I found the study fascinating, I was left a bit confused on how the objectives were framed and felt that I missed a brick in the reasoning. Basically, I'm unsure whether the main goal was:**

We try to clarify below:

- 1) To empirically confirm that sleepwalking episodes can be viewed as dream enactment. This question remains controversial and some authors believe that sleepwalking episodes are automatic and do not reflect dreams (e.g., Tassinari et al., 2009)**

The first aim was to quantitatively evaluate conscious experiences associated with NREM parasomnia episodes. Although prior research has reported dream-like experiences associated with somnambulistic episodes, these findings were based on lifetime recollections, which are theoretically prone to recall bias. To minimize this bias, we conducted interviews with participants immediately after each episode. To our knowledge, only one other previous study has employed a serial interview paradigm to a large number of NREM parasomnia episodes; it was conducted in the 1970s, without a focus on brain activity¹. Our results indicate that, in

many cases, the behavior during parasomnia episodes corresponds to the reported dream content, contradicting the belief that unconsciousness is the norm during sleepwalking. However, we also found that some episodes, notably the brief ones with relatively “stereotyped” behavior, appear to occur without consciousness. Thus, we cannot say that parasomnia episodes always represent ‘dream enactment’. Instead, there appears to be a variability in the level of consciousness associated with parasomnia episodes, although in adults, dream-like experiences are the rule rather than the exception.

The second aim was therefore to provide an explanation for this variability, by relating the degree of consciousness to brain activity patterns measured with high-density EEG. Previous studies that had successfully “captured” and “quantified” brain activity during NREM parasomnia episodes had mostly used expensive or invasive techniques (nuclear imaging, intracranial recordings in epileptic patients) and reported on single cases. Here, by combining high-density EEG with a meticulous cleaning procedure, we were able to analyze regional EEG activity in a comparably large dataset.

In the new version of the manuscript, we completely rewrote the introduction to focus these two aims and provided more context and justification for our objectives.

2) Test the ‘chicken or egg problem’: does the dream cause the arousal or the arousal the dream?

Indeed, we also wondered whether our study could help answer the “chicken and egg problem”. The hypothesis was that if the reported conscious experience was totally unrelated to brain activity during prior sleep, we would only see a difference in brain activity between experience and unconsciousness *during* the episode, but not before. On the other hand, if the episode was related to the prior state of the brain (reflecting dreaming, or the capacity to dream), we would see a difference already *before and also during* the episode.

However, from the reviewer’s comment and after internal discussions we realized that the ‘chicken and egg’ concept was not readily understood and perhaps more confusing than anything. Therefore, in order to simplify the framework of the study and improve clarity, we have omitted this part from the introduction and discussed/interpreted our findings with respect to arousal systems in the discussion section.

I felt that the manuscript subtly oscillated between these two goals and that the analyses and/or results interpretation could change depending on which one is chosen.

- 3. In the case of objective 1 (hyp: sleepwalking = dream-enactment), I feel some discussion points are missing. Indeed, in their Nature Neuro study, the authors computed the neural signatures of dreaming right before the dream report, so supposedly at the very moment when the conscious experience occurred. But here, if one wants to prove that sleepwalkers do dream during the episodes, the main period of interest is the parasomnia episode itself (which, as the authors elegantly say, ‘timestamps’ the experience). However, the EEG analyses computed during the episodes show less robust results than the ones computed before. For example:**

4. The contrast CE vs. NE was associated with lower delta power in posterior cortical regions during the episode, but not exactly in the same regions. This is cool and reassuring. But contrarily to the previous study, there were no differences in beta activity between CE and NE. Why?

Indeed, as the reviewer points out, *during* the episode only low-frequency power, but not high-frequency differed between CE and NE. We can only speculate on why we did not find such a difference also for high frequencies. One possibility relates to methodological aspects: in this study we examined beta power (26-34 Hz) as opposed to gamma power (25-50 Hz) like in our original study on dreaming. This decision was made to limit the contamination of the signal by the many movement artifacts in the gamma range associated with the parasomnia episode. We still found differences in beta power between NE and CE episodes during *prior sleep*, suggesting that there is a continuum in the beta-gamma range as a correlate of conscious experience. However, it is conceivable that that the overall EEG frequency shift associated with the transition from sleep to a state, that at least in adult patients is overall closer to wakefulness, may have resulted in the null findings for the beta range, especially since the first seconds of the episode were excluded, to additionally limit contamination from movement artifacts.

In the current version of the manuscript we have included some of these considerations in the discussion, although they remain tentative.

Moreover, although we used the same methodology as in our previous study on dreaming, because of the movement artifacts associated with parasomnia episodes, we could not analyze the same frequency bands in the beta/gamma range, which may have resulted in some null findings (Figure 2).

Please also refer to our reply to comment 63 of referee 2 who raised a similar issue.

5. The contrast CE vs. CEWR during the episode does not show the same results as the contrast CE vs. NE contrast => in that case, it is difficult to be sure that the conscious experiences reported by patients truly correspond to the episode and not to the dream that happened before (but I agree that we can be more confident when the external behavior matches the report)

We are happy to clarify. The CEWR category helps us dissociate the correlates of the memory of the experience from the correlates of the experience itself, as follows: the contrast CE vs CEWR (forgotten vs. recalled experience) should reflect the correlates of the recall of the experience, while the contrast CEWR vs NE (forgotten experience vs. no experience) should reflect the conscious experience itself, irrespective of the recall of the content. In this study, the contrast CEWR vs NE showed similar results as the contrast CE vs NE, which is reassuring, as they should both reflect the experience. It was expected that the CE vs CEWR contrast would show a different regional correlate, namely that of 'recall', which is the case here, similar to our previous dream study.

We have added additional explanations regarding these distinctions at the beginning to the results section, as follows:

Next we tried to determine the correlates of the recall of the experience by comparing instances of CEWR and CE. Compared to CEWR, CE with recall were preceded, in the pre-episode sleep, by lower delta power and higher beta power in a circumscribed anterior region of the right medial temporal lobe, estimated to comprise the hippocampus, parahippocampal gyrus and amygdala (Fig. 5 left).

- 6. The amplitude of delta waves (+ Slope I and slope II) seems to show an opposite pattern before and during the episode when comparing CE vs. NE (Fig 3 and 4). If higher amplitude (type I) slow waves in fronto-central regions reflect arousal before movement onset, does it mean that patients who report CE are less aroused during the episode than the ones who do not report any experience (NE)? Please discuss.**

The reviewer's observation is accurate: before movement onset, slow waves in frontal regions have *higher* amplitudes and slopes in episodes with reported consciousness compared to those without. Additionally, there are *fewer* slow waves in posterior brain regions. We actually interpret these high-amplitude slow waves (K-complex equivalents/type I slow waves) as a cortical response to phasic activation of subcortical arousal systems, although their functional significance remains unknown. The fact that in episodes without CE, no such cortical 'response' was seen suggests to us that the cortex is less 'responsive' and perhaps not involved in this initial stage of the episode, and that behavior in these cases may be mediated by subcortical structures, similar to what has been shown in animals who display threat or defense related actions without participation of the cortex.

The reviewer is correct that one would expect also more 'type I' slow waves in the front *during* the parasomnia episode. However, it is still unclear whether type I 'equivalents' can persist into the parasomnia episode. In addition, to limit artifactual contamination, we excluded the first four seconds of the episode, where one would most likely expect type I slow wave equivalents. Thus, for us it did not come as a surprise that we did not find a significant difference frontal type I slow wave amplitude *during* the episode.

In the current version of the manuscript, we have already introduced type I slow waves in the introduction, and discussed findings related to type I slow waves with respect to consciousness in the discussion in a dedicated paragraph. We have also provided a schematic figure (Fig. 7 of our findings).

Introduction:

More specifically, compared to reports of unconsciousness, we found that reports of dreaming were preceded by a regional activation of parieto-occipital brain areas (grouped under the name 'posterior hot zone')¹², and in NREM sleep, by high-amplitude frontal slow waves (type I slow waves) that are likely related to arousal systems¹². We hypothesized that if these EEG features reflect core markers of sleep consciousness, they should also distinguish parasomnia episodes with and without report of conscious experience.

Discussion:

Parasomnia experiences were also preceded by large and steep slow waves in frontal and central regions, a slow wave constellation that also precedes reports of NREM dreaming², although in our previous study, it most consistently distinguished experiences with and

without recall of content. These slow waves are reminiscent of so-called type I slow waves/K-complexes which are likely to be related to phasic activations of arousal systems^{3,4} and differ in several aspects from so-called delta waves (type II slow waves) that constitute the “background EEG activity” in slow wave sleep and account for changes in the posterior hot zone². The precise functional and structural correlate of type I slow waves is still unknown. Recently, low-grade activations of the LC in mice were shown to result in an increase delta power in the cortical EEG, while high-grade LC activations induced ‘traditional’ cortical arousal (decrease in low-frequency power and increase in high-frequency power)⁵. It is thus possible that type I slow waves represent a cortical response to such subcortical activations of arousal systems. The fact that the relatively short and stereotyped behaviors without report of consciousness were not associated with this type of cortical ‘response’ could suggest a functional subcortico-cortical disconnection, and possibly a predominantly subcortical generation of behavior in those cases.

And further down:

Taken together, these findings suggest that for individuals to report conscious experiences in relation to parasomnia episodes, 1) posterior cortical regions must be relatively free of (type II) slow waves prior and during the episode, and 2) anterior cortical regions must be in a state in which they respond with type I slow waves to the activation of arousal systems, whether spontaneous or externally induced.

- 7. In general, if the goal was to prove that sleepwalkers are dreaming during the episodes, I am unsure how to interpret the pre-episode EEG signatures of dreaming. Are they showing that the parasomnia episode is simply continuing the dream occurring just before the awakening? Or is the episode a new dream?**

This is a very interesting point. Unfortunately, our results do not allow us to directly answer this question. The fact that parasomnia episodes with report of dreaming can be induced by loud sounds and that the dream contents have similar characteristics across participants (revolving around impending danger), suggests that there is a relation between the activation of arousal systems and the dream content. One could imagine that this ‘arousal’ is integrated into an ongoing dream, or that it leads to the redirection of attention and thus a new dream scene. In the lead author’s clinical experience patients report both possibilities.

- 8. In the case of objective 2 (chicken and egg problem), there are two hypotheses (dream causes arousal or arousal causes dreams). It is unclear how the three specific aims stated at the end of the intro (lines 119-123 p4) will help validate or reject these two hypotheses. Throughout the manuscript, it seems that the main strategy to test the chicken and egg problem is to compare spontaneous vs. provoked episodes. However, it is unclear how exactly this comparison will allow to solve the problem. For example, lines 243-244 is the first mention of an analysis predicting the spontaneous v. provoked nature of the episode as a function of surprise manifestations. But it is unclear until the results section what was the background/hypothesis behind this analysis. What I’m missing is clear predictions spelled out at the end of the intro, such as: if dream causes arousal, we would expect A and B. By contrast, if arousal causes**

dream, we would expect C and D. On an unrelated note, why would arousal and dreams necessarily be related? Could it be that sleepwalkers are dreaming like anybody else and that, at some point, an arousal releases the inhibition of motor outputs, allowing sleepwalkers to enact their ongoing dreams in a similar manner as in patients with RBD?

As discussed in reply to the previous comments, we have simplified the aims in the introduction and dropped the ‘chicken and egg’ formulation.

Regarding the second part of the comment, we thank the reviewer for formulating this additional possibility, namely that the arousal could simply trigger exteriorization of dream-related behaviors without actually contributing to dream content directly. That is of course another possible explanation. However, if we understood it correctly it does not account for episodes without report of consciousness. In other words, if we take the reports of these patients at face value, meaning we assume they sometimes did not experience anything, how can they exteriorize something they did not experience? In addition, if the arousal did not contribute or relate to dream contents and was just a window for dream externalization, we would expect more variability in the dream reports of parasomnia patients. We find it particularly intriguing that the exact same contents are often reported by different patients and that they revolve around ‘impending’ danger. To us this suggests that there is a common stereotyped ‘trigger’ of the contents (activation of arousal systems) which are secondarily contextualized, provided the brain is in a state to contextualize (i.e. dreaming/conscious).

- 9. Finally, if the main question is to investigate the link between arousal, parasomniac episode, and dreaming, additional analyses could be considered. For example:**
- What happens in the dream hot zone when sounds do not trigger an episode?

We agree that this is a very interesting aspect. We have not analyzed these instances, as they were not directly related to the objectives of the present study (comparison CE vs NE). However, we plan to do it in the future as a separate project.

- What happens in the dream hot zone before normal awakening from N3 (i.e., the 25% of the episodes that were not rated as ‘parasomniac episode’?). Why would activation of the hot zone (arousal) provoke the episode? Do you expect such arousal to be more intense than in healthy subjects? If so, could you compare the data in sleepwalkers with the ones from your previous study in healthy subjects?**

The comparison of normal awakenings and parasomnia episodes was the subject of a separate publication ⁶. Compared to parasomnia episodes, normal awakenings out of N3 occurred on a diffusely more ‘activated’ EEG background, up to 60s before movement onset (i.e. consisting in lower delta and higher beta power, Figs. 8 and 9 of the article), as if the awakening progress happened more gradually and started earlier in normal awakenings than in parasomnia episodes. In the posterior hot zone, delta power was lower in normal awakenings compared to parasomnia episodes in the 10s before movement onset. Below we show a hypothetical schematic representation of slow wave activity (type I and type II) before/during parasomnia episodes and normal awakenings. In brief: in the case of NE, patients are unconscious during sleep, the totality of the cortex is inactivated by (type II) slow waves, activation of arousal systems, whether spontaneous or induced, does not lead to large type I slow waves, behavior is observed in the absence of

consciousness. In case of CE, patients are dreaming: posterior cortical regions are consistently free of slow waves but anterior regions are not, anterior regions show high amplitude type I slow waves in response to an activation of arousal systems, behavior occurs and a dream experience is reported. Finally, in normal awakenings, the whole brain wakes up (activates) gradually, but can still generate type I slow waves, waking consciousness (thought-like) is present. Note that reports of experiences were not collected after normal awakenings. As this interpretation is based on data that was not analyzed in the present study, we plan to publish a separate review paper integrating the results of these different studies into a larger theoretical framework.

10. To make a long story short and condensate this long comment (sorry!), could you clarify what the objectives and hypotheses were and discuss the results according to the initial expectations?

We thank the referee for these thoughtful and constructive remarks. We hope that we have adequately addressed them (see replies to comments above).

Minor comment:

To ensure the replicability of the results, please detail the methodology further. In particular:

11. The recovery recording was set between 7.30 and 4pm => it is not a typical time for sleepwalking episodes. Was this time chosen for practical reasons or is there a scientific reason?

The decision to schedule the recovery sleep in the morning (after 25 hours awake) instead of the next evening (after 36-38h awake) was based both on practical considerations, as well as on the results of previous studies.

A previous study with the same inclusion criteria (episodes at least once a month) had shown a threefold increase in the occurrence of parasomnia episodes during recovery sleep after 25 hours of sleep deprivation and sensory stimulation compared with baseline sleep⁷. A threefold increase in the number of parasomnia episodes after 25hours of sleep

deprivation but without sensory stimulation was also reported by another study⁸. One other study reported a 5-fold increase after 38h of sleep deprivation. However, in our experience sleep deprivation beyond 25 hours is very uncomfortable for the participants so we wanted to avoid that.

Although we scrupulously followed the protocol used in the Pilon study, in our study the unanimously rated episodes were not higher in the recovery sleep night (37 episodes) compared to the baseline (38 episodes). In addition, contrary to the Pilon study reporting that 100% of patients presented at least one parasomnia episode during recovery sleep night, in our study 3 patients did not present any episodes.

Since these methodological aspects are a little out of the scope of the current paper, we have not discussed them in the discussion. However, we have mentioned the discrepancy with the Pilon study in a previous manuscript⁶ on a partially overlapping dataset. If the reviewer thinks it is nevertheless important to mention the discrepancy here as well, we'd be happy to include a mention in the results section.

12. L155 p5: How was the 'supervised total sleep deprivation' conducted?

Participants spent the night doing activities of their choice, unrecorded, in one of the rooms of the sleep laboratory. A sleep technician monitored the participants throughout the night to verify that they did not fall asleep. We have added this information in the methods section ('experimental procedure') as follows:

In the evening, they came back for a night of supervised total sleep deprivation in the laboratory, during which they could do activities of their choice in one of the rooms of the sleep laboratory. A sleep technician verified, via continuous video-and audio-recordings, that they did not fall asleep. The following morning, a second hd-EEG (recovery) sleep recording was carried out between ~7.30am and ~4pm.

13. What happens if participants did not wake up once the experimenter calls their name?

When a parasomnia episode occurred, the experimenter waited for it to end before calling the patient's name. If the patient did not react immediately, the experimenter called her/his name a second time. Almost all the time patients responded. However, as indicated in the results section ('Consciousness and parasomnia episodes'), sometimes patients fell asleep very quickly again after the episode. In that case they were not 'awakened' by calling their names and were just left sleeping. These episodes were excluded.

We have added this aspect in the methods section (paragraph 'experimental procedure').

During both the baseline and recovery sleep recordings, when a parasomnia episode occurred, the experimenter waited for the episode to end, called the patient's name and then conducted a semi-structured interview (based on^{11,13}) through intercom about the patient's subjective experiences. If the patient did not react immediately, the experimenter called her/his name a second time. Patients were asked what they had experienced immediately before the examiner had called their name.

14. §L176: end of sentence missing => episode rated as ‘either a parasomnia episode’ or? Please detail the criteria used to define the episodes.

We apologize, it should have read ‘or normal awakening’. We have corrected this mistake.

15. L 177: judgment on specific features associated with the episodes (e.g., interaction with the environment, somniloquia, etc) => please detail the criteria. Were there several, independent judges? If yes, what is the inter-scorer agreement?

There were three independent raters who watched the videos of episodes in a different, random order. Each rater had to score the presence or the absence of behavioral features (interactions with the environment, somniloquia, manifestations of fear and of surprise, rapid onset, startle-like movement sequence, orienting behaviour, and perplexity).

A feature was considered present when at least two of the three judges had rated it as such.

We have added this information to the paragraph in the methods section entitled ‘Rating of parasomnia episodes’ as follows:

Three experts independently rated the behaviour seen on the videos (without the EEG) as either a parasomnia episode (as defined in the ICSD²) or a normal awakening. Only episodes considered as such unanimously by all the three raters were included in the analyses. Experts also rated the presence of the following features: interactions with the environment, somniloquia, manifestations of fear and of surprise (rapid onset, startle-like movement sequence, orienting behaviour, and perplexity), and, when a mental content was reported, whether it was coherent with the behaviour seen on video or not. A feature was considered present when at least two of the three judges had rated it as such.

16. Similarly, how did you judge the coherence between dream report and parasomniac episode? Several methodologies have been used in the literature (e.g., mixing up the dream reports and have a few judges choose the one that matches the behavior the most; have several judges indicate a score of coherence on a scale between 0 to 10, etc). Each method comes with its own biases (no ideal method). Please justify why you choose one approach rather than another.

Raters also evaluated the coherence between report and behavior separately. In case of non-agreement, a discussion between the raters followed until agreement was reached. This method was chosen in order to reduce the complexity of the scoring and to separate coherent and not coherent behavior without reducing the number of trials.

We have added this information to the paragraph in the methods section entitled ‘Rating of parasomnia episodes’ as follows:

Regarding the “coherence between report and behavior”, it was rated separately by the judges. In case of non-agreement, a discussion between the raters followed until agreement was reached. This method was chosen in order to reduce the complexity of the scoring and to separate coherent and not coherent behavior without reducing the number of trials.

17. L188: why use the masseter muscle and not the chin as usually done in sleep studies?

This is how we usually proceed with high-density EEG recordings^{9,10}. In practice, the EEG electrodes overlying the masseter muscles give better signal than those closer to the chin, because of the placement of these electrodes within the high-density EEG cap.

18. L 199: since obtaining clean EEG data in moving participants is a real challenge, the methodology developed here could be very useful for researchers from different fields. Could you elaborate more on the procedure for rejecting movement artifact and provide a figure illustrating the procedure as well as snapshots before/after?

Please find below two images with before and after pictures of the signal at several steps of the procedure. We have included one as supplementary figure (Fig. S8). A more detailed discussion of the procedure is beyond the scope of the current article, but we have written a methodological paper on the application of ASR to sleep (-related) signals that is currently under evaluation by another journal (Somervail et al. Sleep, under revision).

We have also added some information and references regarding the procedure to the methods section, as follows:

ASR is an advantageous method due to its capability to efficiently and automatically eliminate non-stationary artifacts of significant magnitude, irrespective of their distribution on the scalp or their consistency across the dataset¹¹⁻¹⁵.

19. When the patients reported no mental experience, did they remember having moved?

This aspect was not systematically evaluated. Out of the 11 NE, we asked participants six times about whether they had ‘done anything’ or ‘moved’. Five times participants stated that they had not moved and one time the participant reported having “simply” turned around in bed although he had sat up in bed with a very frightened expression. We have added this information as follows:

When asked, most patients said that they were not aware of having displayed any behaviour at all (n=5), while one patient reported ‘simply’ having turned in bed, although he had sat up in bed with a frightened facial expression.

• **Other comments:**

20. The number of episodes seems to be the same during the baseline (n=37) and recovery night (n=38). Does it mean that Pilon’s procedure to trigger episodes is actually not working? One would have expected a lot more episodes during recovery night than at baseline. Given the cost (money- and time-wise) of conducting a full sleep deprivation protocol in sleepwalkers, it might be important to inform the sleep community of the limits of such procedure.

As this comment directly relates to comment 11, we have provided this information in our answer to that comment.

21. Fig 2A: I see no significant cluster between CE and NE during parasomnia episode (delta power), but a significant difference is mentioned in the main text (1358-361).

As indicated by the referee in the next comment, the sentence “Parasomnia episodes with CE were also predicted, compared to those with NE, by lower delta power in posterior cortical regions *during* the episode (Fig. S1A *left*)”, should have referred to Fig S1A *right*. There, at the source level (Fig. 2A right) one can see a significant difference in delta activity between CE and NE episodes. This effect was indeed not seen at the scalp level (Fig S2A right), although the difference goes in the same direction. Discrepancies between source and scalp level statistics like this one can sometimes occur as the procedures used to correct for multiple comparisons are based on different ‘neighbouring’ maps (electrodes/voxels) and because the permutations are newly generated each time. We have clarified in the text that we refer to the lower part of the Figure.

Please note that in the new version of the manuscript, to simplify the presentation of the results, only source level maps are shown in the main manuscript, and results at topoplots level have been moved to the supplementary material.

22. L 360: there is a typo => Fig 2A left should be Fig 2A right

Thank you. This sentence is no longer part of the current manuscript, as the Figures have changed.

23. Please put the corresponding legend below the figure to make readers' job easier and avoid back and forth.

We are sorry that the referee had to go back and forth between legends and figures. We have now incorporated the figures into the text.

Reviewer #2 (Remarks to the Author):

24. In this very interesting study, Cataldi and colleagues: 1) systematically collected parasomnia experiences immediately after spontaneous or provoked NREM-related parasomnia episodes and; 2) compared EEG brain activity immediately before and during episodes with and without conscious experiences. Conscious experiences were reported in 83% (and a specific content in 56%) of the 57 recorded episodes followed by a semi-structured interview (collected from 18 subjects). Compared to reports of "no experience", reports of "experience" were preceded by an activated EEG pattern over posterior cortical regions (lower delta and/or higher beta power). A similar pattern persisted during the episodes when patients displayed complex behaviors and conscious experiences. In addition, content amnesia appeared to be modulated by the degree of right hippocampal activation during prior sleep and the persistence of fronto-parietal slow wave activity during the episodes. These results were interpreted in the light of a previous paper on the neuronal correlates of dreaming, published by the same group. The authors concluded that their findings suggest that parasomnia experiences share the same neural correlates as dreams.

Overall, the manuscript is well written, the aims of the manuscript are clear and logically presented, the list of references is up-to date, the discussion is appropriate, and the statistical analyses appear methodologically sound. However, I have several remarks the authors may want to address before the manuscript could be considered for publication.

We thank the referee for her/his interest in our study and the thorough and constructive feedback.

• Abstract:

25. I recommend to add a statement on the frequency of conscious experiences reported immediately after parasomnia episodes, as this was listed among the main aims in the introduction, and it is actually a very interesting data.

We agree that this is a very important aspect. Following the reviewer's suggestion, we have provided the percentages for episodes with and without conscious experiences already in the abstract.

26. I also recommend to report the specific brain areas found to be significant in the comparison between "experience" vs "no experience" episodes and to talk about the

“dreaming signature” interpretation afterwards, in order to make a clear distinction between results and the conclusions driven by the authors.

Ideally, we would have liked to include, in the abstract, more precise information on the brain activity changes found in association with parasomnia experiences, but unfortunately this was not possible due to the stringent word limitation (150 words) and without losing other information that was essential to the comprehension of the text. We have however added the findings on the posterior hot zone and on type I slow waves. We are aware that the abstract does not reflect all the results but hope that it nevertheless adequately conveys the main findings and what the paper is about.

27. Line 42, page 2: The number of total “potential” episodes was 102, but the number of “sure” episodes was 75, and the number of “sure” episodes followed by an adequate interview was 57. Thus, reporting just the number of 102 may be misleading.

We agree that this may be misleading. We have omitted the number of patients included, so that the sentence remains correct (in fact we recorded more parasomnia episodes than analyzed).

28. Line 49-50, page 2: It is not entirely clear to me how the sentence “and that arousal systems may play a role in the generation of dream contents.” could be justified by the results summarized in the abstract (where slow waves type I and II are not mentioned).

We have omitted this last sentence to accommodate additional results.

The abstract now reads as follows:

Sleepwalking and related parasomnias result from sudden, incomplete awakenings out of slow wave sleep. Clinical observations suggest that behavioral episodes can occur without consciousness and recollection, or in relation to dream-like experiences. To understand what accounts for these differences in consciousness and amnesia, we recorded parasomnia episodes with high-density EEG and interviewed participants immediately afterwards. Compared to reports of unconsciousness (19%), reports of experience (81%) were preceded, during prior sleep, by high-amplitude slow waves in anterior cortical regions and an activation of posterior cortical regions that partially persisted into the episode when patients displayed elaborate behaviours in relation to dream-like scenarios. Amnesia for the experience (25%) was modulated by right hippocampal activation during prior sleep and fronto-parietal slow wave activity during the episode. Thus, the neural correlates of parasomnia experiences are similar to those previously reported for dreams and therefore likely reflect core physiological processes involved in sleep consciousness.

• **Introduction:**

29. It may be worth mentioning among the aims, or as a secondary aim, the comparison between spontaneous and provoked episodes as a function of manifestations of surprise.

Following the comments of reviewer 1, who rightly mentioned that the main focus of the paper was not sufficiently clear, we have rewritten the introduction, focusing on two main aims 1) to quantify conscious experience after parasomnia episodes 2) and to determine what accounts for this variability at the EEG level. The discussion now also reflects this structure: the results pertaining to the main aims are discussed at the beginning, while other aspects, including the relation to arousal systems, provoked and spontaneous nature, are discussed later. We agree with the referee that the spontaneous vs. provoked episode distinction and behavioral aspects (surprise) are an interesting and important aspect. However, given its lower importance with respect to the main aims it may distract from the main message of the paper, we are therefore reluctant to emphasize it in the introduction. Please note that the finding that spontaneous and provoked episodes are behaviorally indistinguishable has been published in our previous paper⁶.

30. Furthermore, there is no mention to the slow wave detection analysis, its background and rational (neither here nor in the discussion but only in the result section).

Thank you for pointing this out. In the new version of the manuscript, both the introduction and the discussion now mention the slow wave findings.

Introduction:

More specifically, compared to reports of unconsciousness, we found that reports of dreaming were preceded by a regional activation of parieto-occipital brain areas (grouped under the name 'posterior hot zone')¹², and in NREM sleep, by high-amplitude frontal slow waves (type I slow waves) that are likely related to arousal systems¹². We hypothesized that if these EEG features reflect core markers of sleep consciousness, they should also distinguish parasomnia episodes with and without report of conscious experience.

Discussion:

Parasomnia experiences were also preceded by large and steep slow waves in frontal and central regions, a slow wave constellation that also precedes reports of NREM dreaming², although in our previous study, it most consistently distinguished experiences with and without recall of content. These slow waves are reminiscent of so-called type I slow waves/K-complexes which are likely to be related to phasic activations of arousal systems^{3,4} and differ in several aspects from so-called delta waves (type II slow waves) that constitute the "background EEG activity" in slow wave sleep and account for changes in the posterior hot zone². The precise functional and structural correlate of type I slow waves is still unknown. Recently, low-grade activations of the LC in mice were shown to result in an increase delta power in the cortical EEG, while high-grade LC activations induced 'traditional' cortical arousal (decrease in low-frequency power and increase in high-frequency power)⁵. It is thus possible that type I slow waves represent a cortical response to such subcortical activations of arousal systems. The fact that the relatively short and stereotyped behaviors without report of consciousness were not associated with this type of cortical 'response' could suggest a functional subcortico-cortical disconnection, and possibly a predominantly subcortical generation of behavior in those cases.

And further down:

Taken together, these findings suggest that for individuals to report conscious experiences in relation to parasomnia episodes, 1) posterior cortical regions must be relatively free of (type II) slow waves prior and during the episode, and 2) anterior cortical regions must be in a state in which they respond with type I slow waves to the activation of arousal systems, whether spontaneous or externally induced.

• **Methods:**

31. If the authors have used the same (or overlapping) dataset of their recently published paper in Sleep, they would need to state this point explicitly here.

The referee is right, 20 of these 22 subjects were included in previous work⁵⁴ comparing parasomnia episodes and normal awakenings. We have added this in the results section as follows:

Twenty-two patients with disorders of arousal were included [14 females, aged 26.9 ± 5.3 yrs (average \pm SD), range 18.3-36.3 yrs, see supplementary material for additional information on patients]. Twenty of these participants were included in previous work⁵⁴.

32. Line 144-146, page 5: How did the authors exclude major psychiatric or neurological comorbidities, sleep apneas with and AHI >15/h, and/or a periodic leg movement index in sleep >15/h?

The presence of major psychiatric or neurological comorbidities was assessed during the clinical consultation, while the presence of sleep apnea and PLMs were assessed with the clinical PSG. With one patient we experienced technical issues during the PSG, so that only the first part of the night was recorded. This participant additionally underwent an ambulatory polygraphy with respiratory and recording of leg movements. Only some parameters of this patient's PSG were included (for instance sleep latency). We have now added this in the text.

All patients underwent a medical evaluation by a board-certified sleep medicine physician and a clinical polysomnography (PSG, see Table S5). One patient experienced technical issues during the PSG and underwent an ambulatory polygraphy with respiratory and recording of leg movements instead. Patients who had presented at least one parasomnia episode during the last month were included. Exclusion criteria were major psychiatric or neurological comorbidities (identified during the medical consultation), medication (except birth control) and pregnancy.

33. I suggest to add, at least in the Supplementary Material, a table summarizing common sleep variables, such as TST, TIB, % and min in each stage, AI, PLMI, AHI.

We thank the referee for this suggestion. We have provided such a table in the supplementary material (Table S5). When reviewing the PSG parameters, we realized that some patients included in this study did not strictly meet the 15/h cutoff criteria for PLMS and AHI. More specifically, two patients had an AHI slightly above 15/h (16.7/h and

15.5/h) and two a PLMS index of >15/h (23/h and 16.4/h). While this cutoff is used in general for our dream studies, we think that in this particular study, it should not have affected results on consciousness specifically, also because the values only slightly exceeded the cutoff. We have revised the methods section accordingly.

34. Line 175, page 6: “while the end was defined as movement cessation”: what if an episode ended in wakefulness, as probably always happened for the analyzed episodes? This is a tricky point. Please provide more details.

Indeed, this is an important issue. The scoring of EEG ‘wakefulness’ vs ‘sleep’ *during* a parasomnia episode is not straightforward, as patterns of sleep and wakefulness can be present simultaneously. In addition, as the reviewer points out, brain activity may evolve or fluctuate within the episode. Therefore, and also because we did not want to ‘bias’ the scoring of behavior, we chose to score parasomnia episodes exclusively based on movement/behavior, irrespective of the EEG. In most cases, at the end of an episode patients displayed a pause in movement, sometimes interacted with the examiner to signal that they were fully awake or resumed a sleeping position again. In the revised version of the manuscript, we have included this information and clarified that the scoring of the awakenings as either a parasomnia episode or a normal awakening was performed independently of the EEG:

“The beginning of an episode/awakening was defined as movement onset on EMG or video, whichever occurred earlier, while the end was defined as movement cessation. In most cases, the end of an episode was marked by either a pause in movement, an interaction with the examiner to signal that patients were fully awake (see for instance hand clapping in video S2) or sometimes the resumption of a sleeping position. Three experts independently rated the behaviour seen on the videos (without the EEG) as either a parasomnia episode (as defined in the ICSD²), or a normal awakening.”

35. Line 195, page 6: regarding artifact subspace reconstruction (ASR) > was this method (and its related parameters) previously validated for sleep? Did you visually inspect each segment after this automatic procedure? May you please provide some examples of the EEG signal before and after the procedure in the Supplementary Material? This may increase the trustworthiness of your procedure.

This methodology was only applied to the period after movement onset, which in most cases was closer to wakefulness than sleep. We used a threshold that allowed us to largely preserve low-frequency oscillations. Most of the segments were visually inspected but not systematically. We now provide screenshots of the signal before and after the procedure in Fig S8 (provided in response to comment 18 of reviewer 1 who raised a similar issue). ASR was not applied to sleep prior to the episode, but in a separate article, which is under evaluation by another journal, we provide a methodological validation of this technique for sleep-related signals (Somervail et al. Sleep, under revision).

Cleaning procedure

36. Line 205, page 7: EEG analyses at the scalp level were performed on the innermost 175 channels to further limit artifact contamination > I may agree, but please provide the readers a rational why this was done only for scalp and not for source analysis.

The reviewer is indeed correct in pointing out that source level analyses were performed on all channels, like in our and other previous publications^{3,9}. This is because in order to perform the source reconstruction, all the channels are necessary to perform the localization of sources, including those on the cheeks.

37. Line 206-207, page 7: “Artifactual channels were visually inspected, marked and interpolated with data from other channels using spherical splines (NetStation, Electrical Geodesic Inc): was this done after ASR? If so, how did you ensure that the reconstructed signal was not reconstructed using artifactual channels?”

Artifactual channels were visually inspected, marked and interpolated after the ASR.

The initial calibration step involves finding relatively clean data to use for reconstruction. Importantly, the algorithm works well if there are some bad channels because the calibration stage is robust against the presence of some noise, a certain amount of bad channels is therefore allowed in the calibration data.

- 38. Line 208, page 7: To exclude ECG and other artifacts, independent component analysis (ICA) was performed > Which other artifacts? Please specify. ICA is a powerful tool, but removing apparently “bad” components (without a clear artifactual meaning) looking only at their topography, may lead to the removal of real EEG signal.**

We agree with the reviewer about the limits of ICA. To limit removal of real EEG signal, like in our previous and other studies, we focused on removing artifacts related to cardiac, ocular, movement and electrodermal activity, based on the topography and the spectrum of the components. We have specified this in the methods section.

To exclude ECG and other artifacts, including ocular, movement and electrodermal activity, independent component analysis (ICA) was performed using EEGLAB routines on sleep and parasomnia episodes separately^{16,17}.

- 39. Line 211-216: were other frequencies computed? If yes, they should be presented in the Supplementary Material. Please, consider also explaining why you selected delta and beta in order to make the rational clear to the readers - who may not always be expert in the topic).**

We have not considered other frequencies than delta and beta power, which we chose as representative of low- and high frequency bands. This choice was made, on the one hand, to simplify the presentation of results, and on the other hand because our analyses were hypothesis driven. In our previous study on dreaming⁹, power in frequency bands between these high- and low-frequency ‘extremes’ (that is: theta, alpha and sigma power), represented very much a grey zone with respect to conscious experiences, in the sense that dream experiences negatively correlated with power in the delta through alpha range and positively with beta power and higher frequencies (unpublished findings). The most consistent findings were however found for delta and gamma power.

The frequency bands were chosen as representative for low-frequency power (slow wave activity) and high-frequency power respectively, based on our previous study on dreaming, and to simplify presentation of results^{15,53}.

- 40. Line 255-256, page 8: The Wald statistics (the squared ratio of the fixed factor estimate over its standard error) were obtained for each model, both with an alpha level of 0.05 and 0.1: I could not see why a threshold 0.1 should be considered here. Please provide a rational, or (better), just avoid mentioning the 0.1 threshold here in Methods and in the Figures/Results.**

Based on our previous study, we had a strong a priori hypothesis about the direction of our results (i.e., we expected lower delta and higher beta power for CE vs NE) and for this reason we decided to use one-sided statistics for the EEG models. In the previous version of the manuscript, we used a 0.1 to emulate the one-tailed statistics (0.1 threshold is

equivalent to using a 0.05 alpha level with a ‘one-sided’ t-test or model). However, in this new version, in order to clarify, we decided to divide the p-values by two and to relabel the thresholds to 0.025 and 0.05. We thought that providing a broader range of significance levels at the topographic level would be informative for the reader, as it helps her/him assess the meaningfulness of the results. For example, the fact that the voxels significant at the 0.05 level are spatially adjacent to the ones significant at the 0.025 level (as opposed to being randomly distributed in the cortex), shows that the results are internally consistent and therefore likely to be meaningful. We have proceeded in this way previously (see for instance Fig. 4C in⁹). We have clarified in the methods section.

In the revised version, we have included the considerations about the a priori hypothesis, and have relabeled the thresholds accordingly.

The Wald statistics (the squared ratio of the fixed factor estimate over its standard error) were obtained for each model, both with an alpha level of 0.025 and 0.05. We opted for the one-sided statistics because we had a priori hypothesis, based on our previous work about the direction of results. In this sense the p-values resulting from the model were therefore divided by two.

- 41. Line 345-348, page 11: “Analyses were based on the 20s of sleep preceding movement onset (“sleep EEG”, in light of our previous studies on the EEG correlates of dreaming), and on the 20s window after movement onset (“parasomnia EEG”, for symmetry and because this timeframe approximately corresponded to the median duration of parasomnia episodes)” > I have one major concern here. You stated that “Episodes for which a clear correspondence between the report and the behaviour was observed were significantly longer than NE (18.8 +/- 12.15 secs vs 37.13 +/- 22.17)”. This might represent a bias when comparing these two conditions. Please either list this issue among the limitations, or to add an analysis on shorter and more comparable segments.**

We thank the reviewer for raising this important point. The choice was made to have a symmetric time frame with respect to the sleep segment for analysis. NE episodes have a shorter duration than CCE, but we considered for analysis the mean power on the time-window for each trial. Considering shorter episodes would have led to exclusion of some trials, which we wanted to avoid. We estimated that 20 seconds were a good compromise of time to analyze. We added this point in the limitations.

Further, parasomnia episodes had a variable duration, but we analyzed a fixed length (up to 20s after movement onset), which may have resulted in less clear-cut findings between categories of consciousness.

- 42. Line 345-348, page 11: “The first four seconds after movement onset were excluded from analysis because of the frequent presence of residual movement artifacts in the EEG.”: are these 4 seconds part of the 20 second window after motor onset (meaning that the analyzed segments were of 16 seconds after all)?**

The reviewer is correct. The analysis window for parasomnia episodes ranges from 4s to 20s after movement onset, resulting in a total duration of 16 seconds. We have clarified this as follows in the new version of the manuscript (text and figure legends):

The first four seconds after movement onset were excluded from analysis because of the frequent presence of residual movement artifacts in the EEG. Thus the analysis for the parasomnia episode was restricted to the timeframe +4 to +20s after movement onset.

• **Results:**

43. Please, if possible, provide more clinical details regarding the patient population (e.g., comorbidities, DoA familiarity, age at DoA onset, DoA subtypes).

We have added this information (see below). We chose to include it the supplementary material (Text S1) as opposed to the results section so as not to disrupt the logic/flow of the paper (as the results section now immediately follows the introduction, to be consistent with Nature Communication guidelines).

Reported age of onset of parasomnia episodes was 9.1 ± 5.6 yrs (3-22). Mean self-reported frequency of parasomnia episodes was distributed as follows: ~once a month: 2 patients (9%), 2-3 times a month: 6 patients (27%); ~once a week: 6 patients (27%); 2-3 times a week 5 patients (23%) and almost every night: 3 patients (14%). All 22 patients (100%) had a history of confusional arousals. In addition, 13 patients (59%) had a history of both sleep terrors and sleepwalking, 6 patients (27%) only of sleepwalking and 2 patients only of sleep terrors (9%). A family history of NREM parasomnias was present in 12 patients (54%). Neurological and psychiatric comorbidities included migraine (n=1), sleep paralysis (n=1) idiopathic hypersomnia (m=1), a history of attention deficit hyperactivity disorder in childhood (n=1) and a history of febrile convulsions in childhood (n=1).

44. Line 334: There was a trend towards a longer duration for CE compared to NE episodes ($p = 0.067$, $z = 1.832$), but not for CE compared to CEWR episodes ($p = 0.546$, $z = 0.603$) (Fig. 1C).> please provide also mean/median values and standard deviations/ranges.

We have provided this information in the Table S3 .

45. There are several repetitions between methods and results regarding the description of the statistical models, I suggest to simplify the text.

The reviewer is correct, some of the methodological considerations were redundant in the results and methods sections. This was partially intended, as wanted to make sure that the reader could understand the results without having to ‘jump forward’ to the methods section, which is presented at the end of the manuscript in Nature Communications. However, the reviewer’s concern is valid. In the revised version of the manuscript, we have removed some of the redundant information.

46. I may have missed this information: was there a difference in the time from onset between episodes with or without mental recall?

Indeed, there was no difference in the “time since lights off” between episodes with or without recall. This information was provided in the fourth paragraph of the results section “Consciousness and parasomnia episodes” (lines 332-334) and in Table S1.

The category of report (CE vs NE or CE vs CEWR) did not vary as a function of the provoked vs. spontaneous nature of the episode or its occurrence during the first vs. second recording or the time since lights off (Table S1).

47. Line 390, page 12: EEG correlates of parasomnia experiences that were coherent and incoherent with behaviour: why there are no source statistics for this specific analysis?

SM did not reveal clear results for delta. As now also discussed in the limitations section, these ‘subgroup’ analyses have limited power. We have added this to the results.

• **Figures**

48. Nose and ear edges seem to be cut out of the print. I suggest to choose better plotrad, intrrad and headrad for figures 2-6 and s1-2.

Thank you, we have provided these changes.

49. Some colorbar units in figures 2-6 are messed up (e.g., W/min). I suggest to make them slightly smaller.

We have changed this according to the reviewer’s suggestion.

50. P values <0.1 have no statistical meaning. I suggest to avoid plotting these values from figures 1-6 and s1-2.

Please see our reply to comment 40 regarding this point.

51. The blue bar representing sleep at the top of many plots in Figures 2-6 and s1-2 is slightly asymmetric compared to the mvt onset and between plots.

The asymmetry between the bars was intended to represent the excluded first 4s of the parasomnia episode. However, following the reviewer’s comments, we have uniformed the bar across figures and reduced the differences in bar length between the sleep and parasomnia bars.

• **Language:**

52. I recommend the authors to go through the text accurately and make minor stylistic changes and grammatical/punctuation corrections. Below, only a few examples.

See reply to specific comments below.

53. Line 74, page 3: there is a space between the word “consciousness” and references 1-2, while for subsequent references no space is used > please revise the refences according to the Journal guidelines.

This was corrected.

54. Line 89, page 3: I am not sure why you used square brackets without content [...], in the middle of this long statement.

The ellipsis (three points between brackets) referred to a part of the original citation that has been omitted. This sentence is however no longer part of the manuscript.

55. Line 94, page 3: For one > First.

This sentence is no longer part of the manuscript.

56. Line 110, page 4: In the present study: If starting the sentence with a prepositional phrase, then a comma should separate the phrase from the rest of the sentence.

This sentence is no longer part of the manuscript.

57. Line 122, page 4: “EEG signatures of dreaming 3) to clarify whether the” > a comma is missing here. Consider the use of semicolons instead of commas.

This sentence is no longer part of the manuscript.

58. Line 145, page 5: medication > medications.

If the reviewer is not opposed, we prefer to keep ‘medication’ in the singular form as this formulation is more commonly used.

59. Line 175-177, page 6: Three experts independently rated the behaviour seen on the videos as either a parasomnia episode (as defined in the ICSD2). > Perhaps an “or” is missing here.

Absolutely. It should have read ‘or a normal awakening’. This was corrected.

60. Line 197, page 6: please use “that” in defining clauses, “which” in non-defining clauses (preceded by a comma)

This was corrected.

61. Line 229, page 7: algortihm > algorithm

This was corrected.

• **Discussion :**

62. Line 450-452, page 14: “a previously documented electrophysiological pattern that was found to distinguish reports of dreaming from those of no experience across REM and NREM sleep”: looking at the previous paper, I am not fully convinced that the brain regions are exactly the same, especially regarding beta power, but it is not easy to understand from the figures. Maybe you could specify overlapping and not overlapping regions (if any).

To more directly compare the findings in the current study and our previous study, we have computed conjunction maps combining the results of the contrast CE-NE from our current and previous studies (experiments 1 and 2 in⁹). We provide these figures now in the supplementary material (Figure S2). For the sleep before the episode, the overlap with our previous study centers mainly on primary visual areas, while for the episode, the overlap centers mainly on higher order (associative) areas, including the left precuneus, the superior parietal lobule, the supramarginal and angular gyri and posterior cingulate. As the reviewer noted, while there is clear overlap, the results are not exactly identical between our current and previous study. Indeed, some variability was expected, as the hot zone is relatively large and we also found some degree of variability between the two dream studies. Please also note that for the high-frequency power differences, the results are not directly comparable between the studies, as in our previous study we analyzed the gamma band, and in the current study the beta band. The statistics were also not computed in the same manner (paired t-tests in the previous studies and generalized linear mixed models in the current study).

Fig. S2:Conjunction maps: differences and overlap between contrast conscious experience/no experience in the current study and the contrast dream experience/no experience in two previous experiments on the neural correlates of dreaming in healthy participants (Siclari et al., Nat Neurosci 2017) for low frequency power (A) and high-frequency power (B). Power in the high-frequencies was analyzed in the beta band in the current study (26-34 Hz), and in the gamma band (20-50Hz) in the previous study on dreaming. Statistics were not computed in the same manner (generalized linear mixed models in the current study, paired t-tests in the dream experiments).

63. Line 452-455, page 14: “Parasomnia experiences that were coherent with the observed behaviour not only displayed the ‘dream signature’ during prior sleep, but also during the episode compared to NE (in the form of reduced delta power in

posterior cortical regions), as if the brain remained in ‘dream mode’ after the initial arousal.”: This is very interesting, however, I have two questions: 1) How do you explain the difference between the pattern observed prior to motor onset and the pattern after motor onset (clearly more anterior for delta power?); 2) where are the statistics/figures for the source analysis?

- 1) The areas displaying significant NE-CE differences during prior sleep and during the episode *both* overlapped with the posterior hot zone identified in our previous study. However, the differences *before* the episode were localized mainly to primary and secondary *visual* areas, while *during* the episode they involved more parietal, higher-order, and in part multisensory associative areas including the precuneus, the supramarginal gyrus and the angular gyrus. One possible explanation for this difference may relate to the type of experience. During sleep patients are largely disconnected from the environment, whereas during parasomnia episodes they have their eyes open, and by definition move and interact with their environment. The fact that during the episodes, CE differed from NE in more parietal and somatosensory-related areas could reflect that conscious experiences more consistently had a sensory component during episodes than during the preceding sleep. We have added these considerations to the discussion, acknowledging their speculative nature.

Interestingly, while before the episode differences between CE and NE were mainly localized to primary and secondary visual areas, during the episode they involved more parietal, higher-order, and multisensory associative areas. It is not clear what underlies these differences in topography, but contrary to sleep, during parasomnia episodes, by definition, patients moved around and interacted with their environment. One tentative explanation is therefore that the involvement of higher-order areas during the episode reflects a more consistent (multi)sensory component of experiences compared to the preceding sleep.

- 2) Regarding statistics: the contrasts shown at the source level in 2A and B reflect voxels for which the Wald statistics were significant.

64. Line 478-479, page 15: “In conclusion, somnambulistic experiences not only share core features of dreams, but also display similar neurophysiological correlates as dreams.” > Perhaps, as you previously suggested that the ‘posterior hot zone’ constitute a core correlate of conscious experiences in sleep, what is mainly shared here, is the fact that patients are conscious before and during episodes. Although, yes, this does not completely fit with the fact that during episodes the posterior hot zone seems slightly more anterior (see previous point).

Thank you for this suggestion. We have added this consideration in the abstract (as outlined previously) and in the discussion (as follows):

Taken together, our results suggest that not only do NREM parasomnia experiences display the core features of dreams, but they are also associated with similar brain activity patterns, which are therefore likely to reflect fundamental neurophysiological mechanisms involved in sleep consciousness¹⁸.

65. If I understood correctly, in your previous paper, the EEG activity during episodes and during physiological movements in patients did not differ. Did you collect mental content after physiological movements? What would you expect? Could you please comment on this point?

In our previous study ⁶ we indeed compared N3 awakenings scored as normal awakenings with those scored as parasomnia episodes. When only considering unanimously rated parasomnia episodes, like in the current study, we found that normal awakenings displayed lower beta and delta power after movement onset (Figure 9 of the manuscript). We did not collect mental content after the normal awakenings. We devised a simplified, hypothetical schematic figure representing ‘what we would expect’ in terms of consciousness and corresponding slow wave constellation before/during the different types of awakenings, which is provided in response to comment 9 of referee 1 who raised a similar issue. As this interpretation is based on data that was not included in the present study and is partly hypothetical (no reports were collected after normal awakenings), we plan to publish a separate review paper integrating the results of these different studies into a larger theoretical framework.

66. Existing controversial points (e.g., other findings about neural correlates of dreams by other groups), limitations and future research needs could be discussed more thoroughly.

We have included this information in the paragraph ‘limitations’ as follows:

Finally, while our results are concordant with those of one of the largest studies on the EEG correlates of dreaming that used the same methodology, these results still need to be confirmed by other studies of the same type.

Reviewer #3 (Remarks to the Author):

Review of Non-REM parasomnia experiences are associated with EEG signatures of dreaming

In this article, Cataldi et al. explore the neural correlates of NREM parasomnia events, and the content of these states (i.e. dreaming, no-dreaming). They further attempt to classify whether the behaviours during a parasomnia is associated with dream content. While an interesting topic, I have some reservations regarding the setup, the analysis and the interpretation of these results, as specified below.

We thank the referee for the thorough review of our manuscript and for the constructive and insightful comments.

Introduction:

67. I like the inclusion of ‘pop’ Shakespearean topics, but I am sure there is an actual reference to sleepwalking that would be more pertinent to a science journal.

The reference to Shakespeare at the beginning of the paper was included to illustrate that the question of whether a sleepwalker is conscious is not only of interest to scientists or physicians but has also been formulated in other fields. We thought that this was particularly relevant for a journal like Nature Communications with a broad readership. Following several of the referee's comments on the introduction, as well as the comments of the two other referees, we have however completely rewritten the introduction to provide a more focused presentation of the aims. The reference to Shakespeare is no longer part of the manuscript.

68. If you are going to call sleepwalking by it's common name, keep it, or at least make reference to the fact that it means 'somnambulism.'

Thank you for pointing out this issue. We have clarified that sleepwalking means somnambulism in the first sentence of the manuscript:

Sleepwalking (somnambulism) is an enigmatic condition characterized by sudden but incomplete awakenings out of Non rapid eye movement (NREM) sleep, during which affected individuals may interact with their environment in an altered state of consciousness^{1,2}.

69. 1960's? I don't think I have ever seen it written 1960ies.

We changed "1960ies" to "1960's".

70. The debate of whether or not dreaming occurs during sleeping, or is a confabulation, is interesting, but it doesn't really have to do with the article at hand. Like you say at the beginning of the second paragraph, there is no more doubt about whether dreams happen in NREM sleep. This whole debate doesn't set up anything to do with your article.

Following the reviewer's comment, we have omitted this 'debate' from the introduction.

71. Page 3, line 97-98, maybe rephrase, difficult to parse. It seems like you're saying that activity related to some event is present during sleep, and during wake, and therefore we can say they reflect the same experience. But the whole point of neuroimaging is that it is correlational, you can't draw causation from such things.

The referee refers to the following sentence « Second, advances in signal analysis and acquisition have allowed researchers to pinpoint brain activity patterns during sleep that not only correlate with reported dream contents, but also with neural activations that are seen when these same contents are perceived during wakefulness, suggesting that dream reports can be taken at face value to reflect experiences that occur during sleep¹⁴⁻¹⁹. »

Following the revision of the introduction, this sentence is no longer part of the manuscript. However, we appreciate the reviewer's request to take this issue into account and take the opportunity to comment on the way we interpret our findings in general. While we agree that it is impossible to 'prove' causation, i.e. that someone actually experienced a conscious content (for example a face) even when (s)he reports so and there is an activation in the corresponding brain area (i.e. the fusiform face area) prior to

the report, in our approach we take subjective reports at face value. In fact, we think that the large majority of delayed reports should be taken at face value until proven wrong rather than the other way around, as a straightforward consequence of scientific abduction, i.e. inference to the best explanation. As also stated by Windt^{19,20} in her treatment of the scientific study of dreaming sleep, we think that while exceptions are important (and should be investigated systematically), overall researchers should take dream reports at face value (both for what is reported and what is not reported) when they are collected in an appropriate way under optimal laboratory conditions.

72. Line 103-104; same deal; EEG hallmarks of sleep occur while people are awake, and therefore these must relate to sleep? This is again drawing causation from a correlative measure. I am not saying that these parasomnias aren't sleeping states, but placing the argument as, they are awake, but have sleep signals in the brain, so these co-occur doesn't really make sense to me. Could it be that these people are asleep? This whole thing seems rooted in a debate on the nature of 'awake' and consciousness more generally. I'd avoid the whole debate.

We realize we have not been sufficiently clear in this introductory paragraph. When referring to studies that showed EEG patterns similar to both sleep and wakefulness in different brain regions during parasomnia episodes, we did not mean to conclude on whether patients with parasomnia were actually asleep or awake, as parasomnias likely represent a behavioral state that cannot easily be classified as sleep or wake. Instead, we tried to make clear that this variability in regional brain activity patterns is an interesting resource to try and understand how it relates to conscious experiences. We have now rewritten the introduction and hope that it is clearer.

73. SWS? Please define acronym.

We have omitted the abbreviation of "slow wave sleep" and written it out since it only occurred three times in the manuscript.

74. For most of your analysis steps, the choices seemed pretty standard. I was curious what the purpose of the slow wave analysis was, specifically, and why so many different parameters were extracted. Do you have a theory as to why each of these would be interesting, perhaps correlating with something to do with sleep? Otherwise it seems like fishing...

The different slow wave parameters and their topography allow us to distinguish between two types of slow waves that 'behave' very differently and therefore likely have different functional correlates. For instance, using this channel-by-channel slow wave analysis, we have previously shown that the two types of slow waves display distinct variations across the night⁴, occur on different EEG backgrounds⁴, induce specific EEG changes⁴, are differentially affected by development²¹ and experience²², and importantly, bear an opposite relation to dreaming. In the present case, the power analysis revealed delta power differences in posterior cortical regions between NE and CE during the sleep before the episode. The subsequent slow wave analyses showed that this difference is likely related to differences in slow wave density in posterior cortical regions (as we have previously shown), while slow wave amplitude is paradoxically *higher* in frontal regions in CE compared to NE. The functional correlate of these high-amplitude slow waves is currently

unknown, but we prefer to mention their presence before episodes with CE. Otherwise, from the power analysis one could get the ‘false’ impression that slow waves only differ in the back of the brain. However, we realize that not all slow wave parameters may be not so informative. In the new version of the text, we have limited our presentation of slow wave parameters to only the most important ones and included the others/all of them in the supplementary material.

- 75. The slow wave analysis could be described in more detail. With the limited work I have completed with this (a few years ago) using the WISPIC toolbox, there are more arbitrary choices for the definition of a slow wave that are not sufficiently reported. It is also unclear what this adds over delta power, my impression is that it does not add much.**

We used a custom made MATLAB script based on a script developed originally at the WISPIC (Ref Brady), although it is not the one of the WISPIC toolbox. In brief, the choices we made related to filtering (0.5-4Hz; stop-band at 0.1 and 10 Hz using a Chebyshev Type II filter), down sampling (from 500 to 128 Hz), duration threshold (only half-waves with a duration between 0.25s and 1s were considered), not including an amplitude threshold, and length of the moving average filter (50ms) for the derivative.

We have specified this in the methods section as follows:

A previously validated slow wave detection algorithm^{12,62} was applied to the EEG signal. The EEG signal was baseline corrected, subtracting to each electrode signal its mean. Then it was re-referenced to the average of the two mastoids electrodes, downsampled to 128 Hz and bandpass filtered (0.5-4Hz; stop-band at 0.1 and 10 Hz) using a Chebyshev Type II filter (Matlab, Mathworks). The slow wave detection was performed for each channel separately and consisted in identifying the zero crossings, the negative deflections between two zero crossings were defined half-waves. Only half-waves with a duration between 0.25s and 1s were considered, no amplitude threshold was applied. After applying a moving average filter of 50ms, the determination of negative and positive peaks was based on the zero crossings of the derivative of the signal. Then for each wave different parameters were extracted: maximum negative peak amplitude, slope 1 (between the first zero crossing and the negative peak), slope 2 (between the negative peak and the second zero crossing), the number of negative peaks and the duration (time from the first and second crossing). Finally, the parameters of the waves contained by each instance were averaged and the density (expressed in number of slow waves per minute) was extracted.

- 76. Stats, was it CE vs (NE or CEWR) or CE vs NE vs CRWR? Surely CEWR and CE would be more similar as a grouping than NE and CEWR. Please report your formula for GLMMs somewhere. A simple description doesn’t really cut it; for example, what are main effects, what are included as interactions etc etc. 0.1 is a high cluster inclusion alpha. Usually, I’d use 0.05 or 0.025.**

We are happy to clarify. CE, NE and CEWR were always compared separately and never grouped. Fig 2 and 3 show the contrast CE vs. NE, Figs 5 and 6 CE vs. CEWR and Fig. S3 CEWR vs. NE. Then the CE set was subdivided into CCE (parasomnia episodes with clear coherence between the behavior and the report of conscious experience) and ICE (parasomnia episodes with *not clear* coherence between and report of conscious

experience). Fig. 4 shows the contrasts CCE vs. NE and ICE vs. NE. The GLMMS formulas are reported in Tables S1 and S4.

Regarding the choice of our statistical threshold: based on our previous study, we had a strong a priori hypothesis about the direction of our results (i.e., we expected lower delta and higher beta power for CE vs NE) and for this reason we decided to use one-sided statistics for the EEG models. In the previous version of the manuscript, we used a 0.1 to emulate the one-tailed statistics (0.1 threshold is equivalent to using a 0.05 alpha level with a ‘one-sided’ t-test or model). However, in this new version, in order to clarify, we decided to divide the p-values by two and to relabel the thresholds to 0.025 and 0.05. We thought that providing a broader range of significance levels at the topographic level would be informative for the reader, as it helps her/him assess the meaningfulness of the results. For example, the fact that the voxels significant at the 0.05 level are spatially adjacent to the ones significant at the 0.025 level (as opposed to being randomly distributed in the cortex), shows that the results are internally consistent and therefore likely to be meaningful. We have proceeded in this way previously (see for instance Fig. 4C in⁹). We have clarified in the methods section as follows:

The Wald statistics (the squared ratio of the fixed factor estimate over its standard error) were obtained for each model, both with an alpha level of 0.025 and 0.05. We opted for the one-sided statistics because we had a priori hypothesis, based on our previous work about the direction of results. In this sense the p-values resulting from the model were therefore divided by two.

77. It sounds like you did a cluster-based permutation test. Did you use a function for this, or calculate it all manually? How did you define clusters (temporally, spatially, both?), you mentioned electrodes, but how did you decide on neighbouring electrodes? For example, I’d normally use fieldtrip to get a list of neighbouring electrodes based on distance.

The referee is correct, a cluster-based permutation test was applied to the statistics obtained by GLMM models, using 1000 iterations on dummy populations obtained by shuffling the labels at each iteration. The script was written manually written in R. Electrodes/voxels clusters were defined spatially, and neighbors were defined through a fieldtrip function, using the triangulation-method, which calculates a triangulation based on a two-dimensional projection of the sensor position. We have added this information to the manuscript.

For each permutation, the model was applied and neighbouring electrodes with p-values < 0.05 or 0.1 (according to one or two-tailed statistics) were identified as a cluster. Electrodes/voxels clusters were defined spatially, and neighbors were defined through a function from the matlab toolbox fieldtrip, using the triangulation-method, which calculates a triangulation based on a two-dimensional projection of the sensor position.

Results

78. How did you confirm that the unprovoked parasomnia events were unprovoked? Were the rooms sound proof etc etc.

The recordings took indeed place in a certified sleep center, guaranteeing conditions that are appropriate for sleep recordings, including noise levels etc. We have added this information in the paragraph 'EEG recordings' as follows:

Sleep was recorded with a hd-EEG system (256 channels, Electrical Geodesics, Inc., Eugene, Oregon) and a 500Hz sampling rate, in a certified sleep center, guaranteeing conditions that are appropriate for sleep recordings.

We are confident that the spontaneous parasomnia episodes were not induced by ambient sounds because the experimenter monitored the participants continuously through video and audio and would have noticed the occurrence of sounds unrelated to the study setup. In addition, all the video recordings, including audio, were reviewed by the experimenter retrospectively as well. Finally, it should be noted that the sounds that were successful in provoking parasomnia episodes were often very loud, it is thus unlikely that such a strong spontaneous sensory stimulation would have gone unnoticed.

79. You note that the types of episodes are behaviourally indistinguishable. Is it possible that the two types of events are both caused by some type of stimulation, however we just don't know the cause of the 'spontaneous' awakenings?

As outlined in the reply to the comment above, it is highly unlikely that an environmental sensory stimulation, strong enough to provoke a parasomnia episode, went unnoticed by the examiner. However, we do think that both spontaneous and provoked episodes are induced by an activation of the same 'arousal systems'. Spontaneous, recurrent activations of arousal systems are an integral part of Non-REM sleep, as recently confirmed by studies demonstrating periodic activations of the locus coeruleus^{23,24}. It is therefore not too far-fetched to assume that spontaneous activations of arousal systems, albeit with a pathological timing intensity or motor coupling, could underlie the occurrence of NREM parasomnia episodes.

We have formulated these considerations now more explicitly in the discussion (second paragraph):

The fact that almost identical behaviours, EEG patterns⁶ and mental contents are observed between spontaneous and provoked parasomnia episodes supports the hypothesis that provoked episodes mimic a naturally occurring arousal process²⁵. Indeed, recurrent activations of arousal systems are now known to be an integral part of NREM sleep^{26,27}, as recently confirmed by studies demonstrating periodic activations of the locus coeruleus in slow wave sleep^{23,24}. It is therefore not too far-fetched to assume that spontaneous, naturally occurring activations of arousal systems, albeit with a pathological timing⁶, intensity and/or motor coupling, could underlie the occurrence of NREM parasomnia episodes.

80. Line 334; there wasn't a trend; p-values are an (arbitrary) cutoff we use in science, things are either significant or not significant.

Median durations of CE episodes were longer than CEWR and NE episodes, (see table S1) but these contrasts did not reach statistical significance (CE vs. CEWR: $p = 0.546$, $z = 0.603$ and CE vs. NE: $p = 0.067$, $z = 1.832$).

81. Did you look at the beta/delta ratio for eeg activity? Ratio of high to low frequency activity is pretty well established in, sleep (and your prior) research.

We have not performed this analysis. In fact, the two frequency bands are known to be anticorrelated during sleep; however it was not clear whether this would be the case during parasomnia episodes. For this reason, we considered the two frequency bands separately.

82. Line 370-377: this is the type of context that would have been helpful above where I asked why you looked at these various slow-wave effects.

Thank you. We have now provided information on slow waves already in the introduction section.

83. For all these EEG stuff, where are the stats? I see in the figures you gave us p-value stats, but a) you should be giving cluster statistics as well, and b) if you are doing a cluster-based permutation analysis, you should be reporting a p-value for a given cluster, rather than individual electrodes (your image could be confused in this way). Plus, you should report precise p-values unless $p < .0001$ or so; $p < .05$ is something out of the 80s.

The statistics are represented as follows: the colors of the topoplots indicate the precise t-values (on a scale from -4 to 4), whereas the white and black dots denote the channels for which the contrast was significant (with two different p-value thresholds, respectively) after the cluster-based permutation analysis used to correct for multiple comparisons.

We have added the cluster statistics as a supplementary table (Table S4). To simplify presentation of results, only the cluster statistics supporting the main conclusions (spectral power) were reported. Indeed, including the cluster statistics for all the slow wave analyses would add a large amount of data because of the many parameters and time frames considered and would likely make the table not very reader friendly. However, if the referee retains that it is of interest, we would be happy to provide this information in the supplementary material.

84. Line 400- 404; you could test for this statistically. If you don't, drop the line. Causal inference from one thing not happening is really weak. Also, as an aside, because using negative inference as you have done here begs the question; what kind of power did you have for this study?

We agree with the reviewer that we cannot ascertain that this particular subtype analysis had enough power to draw this conclusion. We have dropped the line in question from the results and have discussed this aspect in the limitations.

85. I think a major issue is whether 22 people, with 107 events to analyze, was enough to find a difference in the perceived behaviour reported by the technician analyzing said behaviour and the (obviously imperfect) report of a patient who just woke up from a parasomnia event? This is very important given the heterogeneity the authors emphasize.

This reviewer rightly points out a number of methodological challenges inherent to the study of parasomnia episodes. Below we outline how we addressed or limited the issues raised:

- *Sample size*: although 102 events may not seem many, they should be seen in the context of other existing papers quantifying brain activity associated with NREM parasomnia episodes, some of which were highly influential, and which included mostly single episodes or patients^{28,29}. Here we have invested a considerable amount of work in recording many episodes and in developing an artifact removal procedure allowing us to quantify EEG activity associated with parasomnia episodes. Not only did this sample size allow us to find significant topographical differences between CE and NE, but these differences were also in line with our previous findings and hypothesis, increasing our confidence in the meaningfulness of the results. However, we agree, as stated in our reply to the previous comment, that for some subgroup analyses (coherent/incoherent episodes) we may not have had sufficient power to draw firm conclusions. We have now discussed this in the limitation section.
- *Validity of scoring parasomnia episodes by a technician*: to ensure validity of scoring of parasomnia episodes, in addition to the highly experienced technician, two other experts reviewed the videos independently and in a different order. Only episodes that were unanimously scored as such by the three raters were included in the EEG analysis. This way of proceeding should have ensured consistency and validity of the scoring of parasomnia episodes.
- *Validity of patients reports*: as also outlined in our response to comment 71, here we adopted the approach to take participants' reports of conscious experience at face value, as the best inference to scientific explanation. By doing so we found anatomically plausible and distinct EEG correlates for the recall of experience and the experience itself. The latter were largely in line with those found in our previous study in healthy participants, suggesting that reports given by patients, even after a parasomnia episode, reflect similar neurophysiological processes associated with dreaming.

86. In this results section you have done a lot of 'trying to predict x from y' type of analysis. Did you consider perhaps a machine learning approach, rather than GLMMs?

We have used the terminology because the models we used were indeed 'predictive' models. Machine learning is a good method to find what best predicts a phenomenon, however in our case, the specific GLMM models were driven by specific questions. Our intention was not to find the best predictor of a phenomenon but to understand how it relates to variables of interest.

87. How much do you trust deep structure analysis like hippocampal and amygdala reconstruction from EEG? If each patient didn't have an MRI (and really, probably even if they did) using a generic head model is probably going to lead to unreliable results in this way. You can get 'ghost' deep sources due to the math of how source

reconstruction works (electrical spreading etc.).

Thank you for pointing out this issue, which is relevant to studies using high-density EEG recording and source localization. We are aware of these limitations and of the question of whether these techniques can adequately image deeper structures like the hippocampus. Despite these limitations, we think that these techniques are suited to answer the questions assessed by this study. High-density EEG and source modeling has allowed us, in the past, to find highly plausible and localized perceptual correlates that were shared across sleep and wake (i.e. face fusiform area for faces seen in a dream), increasing our ‘trust’ in these methods. Regarding deep structures, recent papers directly comparing scalp EEG with intracranial EEG showed remarkable convergence in the localization of deep source activity including the hippocampus^{30,31}. In addition, in the present study the hippocampal activation was also anatomically plausible, given that it distinguished reports of recall from those with no recall. Please note that while we did not use an individualized head model based on MRI, we used individualized geocoordinates of the electrodes, obtained following a procedure implemented in GeoSource 3.0 (NetStation, Electrical Geodesics, Inc). We agree with the reviewer that caution is warranted. Therefore, when describing the results relating to hippocampal activation, we have used a more careful formulation:

Compared to CEWR, CE with recall were preceded, in the pre-episode sleep, by lower delta power and higher beta power in a circumscribed anterior region of the right medial temporal lobe, estimated to comprise the hippocampus, parahippocampal gyrus and amygdala (Fig. 5 left).

Discussion

88. 56% as consistent with the ‘rule’ rather than the exception isn’t really a strong rule.

We agree that our sentence may not have reflected results in the best way. We have rephrased the sentence in question to better reflect our results. It now reads:

Patients’ reports ranged from no or minimal experiences to dreamlike scenarios characterized by delusional thinking, multisensory hallucinations and volitional interactions with the environment. Patients reported conscious experiences after 81% of episodes, could clearly recall the content in 56% and reported no experience in 19%.

89. Line 460- going from activation of the arousal system to an interpretation of a danger signal might be right; but if you are arguing that the dreams are online rather than post-hoc confabulations; how can we tell? Maybe we get a strong arousal signal, and then when we wake up, we have to make up a reason for why we had this arousal? Like the classic neurostimulation experiment of the girl who would laugh if her SMA was stimulated, and would, post-hoc, make up reasons for why this was.

The referee is correct. Following this interpretation, arousal (= activation of arousal systems) could be a driving factor in creating a dream content in parasomnias, rather than the other way around (that the dream creates the arousal). The fact that dream contents are so similar across patients and parasomnia experiences (as also previously reported in the literature), revolving around (impending) threat, would sustain such a hypothesis. In addition, parasomnia episodes with such mental content can be induced by arousing

stimuli, like in the present study, provided that certain background conditions are met (posterior hot zone not deactivated and type I slow wave 'inducible' in the front of the brain). Further, episodes started in a similar way but became more complex idiosyncratic over time, suggesting a secondary unfolding of the episode/dream and elaboration of an initial stimulus.

Given that activations of arousal systems are an integral part of normal sleep (even without inducing full awakenings), and that the "signature" of normal and parasomnia 'dreaming' is similar and reflects activation of arousal systems, one could hypothesize that a similar mechanism underlies NREM dreaming in general, or perhaps at some types of NREM dreams. We have now discussed this more clearly in the last part of the discussion.

Our results also raise the intriguing possibility that arousal systems contribute to the generation of dream contents. In this study we could induce parasomnia episodes with loud arousing stimuli and observed that reported dream contents, as also described previously, thematically often related to impending danger, suggesting a relation between the arousing stimulus and the dream content. Further, episodes started in a similar way across patients but became more complex and idiosyncratic over time, suggesting a secondary elaboration or interpretation of an initial stimulus. It is tempting to assume that a sudden activation of arousal systems, whether induced or occurring spontaneously, is interpreted as a danger signal by patients and secondarily contextualized, provided that the brain is in "dream mode/conscious" prior to the arousal, remains in this state after the arousal and thus "can make something" of it. Whether such a mechanism could also underlie (NREM) dreaming in general, or perhaps certain types of dreams, remains to be elucidated. On the other hand, when behavioral arousal occurs in a state that is not compatible with dreaming/sleep consciousness (cortex inactivated by type II slow waves), it is not associated with experience.

90. Do you think all this analysis of EEG was warranted, given the limited scope of what you report in the discussion? I would focus on the power analyses.

We understand that the amount of analyses included may be overwhelming for the reader, and that the power analyses would have been sufficient to make the main points in the paper. On the other hand, with respect to power analyses showing posterior cortical activation as a correlate of NREM parasomnia experiences, the slow wave analyses show, like our previous study on dreaming, that frontal brain activity also differs as a function of consciousness in parasomnia episodes, although the functional significance of this frontal brain activity, which is likely arousal-related but looks 'sleep-like' (type I slow waves) still remains unclear. We hope that these observations can stimulate future research studies to understand the functional significance of these EEG features. Therefore, if the reviewer is not opposed, we would like to maintain this analysis in the manuscript. To improve readability of the manuscript, we have simplified the presentation of results. We now only show the source statistics in the main figures and have limited our presentation of slow wave parameters to only the most important ones.

91. 477- why do you keep using the phrase 'somnambulistic;' as you report here, no one left their bed during the experiment.

We have omitted the term somnambulistic.

92. 479 – so your answer to the chicken and egg question is, why not both? Maybe a different analogy is needed. That the effect is somewhat cyclical or builds up makes intuitive sense though.

Indeed, we realized that the chicken and egg analogy is confusing and have taken it out from the manuscript.

93. This final paragraph makes much more sense and clarifies what you're trying to say. Basically, you're reporting there is a potentiation, and then there is an arousal. If the potential is low, a large arousal would be needed to generate a parasomnia, but people might wakeup first. If the potential is high enough, even an indiscernible arousal could cause a parasomnia event. Maybe make this clearer earlier on, however this is unlikely to be unique to parasomnia and hence I think the unique insights for parasomnia are limited.

We also think that our hypothesis is not specific to NREM parasomnias but could apply to NREM dreaming in general. We have now included a schematic figure to illustrate our results (Fig. 7) and hope this helps the reader to better understand them.

References

1. Fisher, C., Kahn, E., Edwards, E., Davis, M. D. & Fine, J. A psychophysiological study of nightmares and night terrors. III. Mental content and recall of stage 4 night terrors. *J Nerv Ment Dis* **158**, 174–88 (1974).
2. Siclari, F., Bernardi, G., Cataldi, J. & Tononi, G. Dreaming in NREM sleep: A high-density EEG study of slow waves and spindles. *J. Neurosci.* **38**, 9175–9185 (2018).
3. Siclari, F., Bernardi, G., Riedner, B. A., LaRocque, J. J., Benca, R. M. & Tononi, G. Two Distinct Synchronization Processes in the Transition to Sleep: A High-Density Electroencephalographic Study. *Sleep* **37**, 1621–1637 (2014).
4. Bernardi, G., Siclari, F., Handjaras, G., Riedner, B. A. & Tononi, G. Local and widespread slow waves in stable NREM sleep: Evidence for distinct regulation mechanisms. *Front. Hum. Neurosci.* **12**, 1–13 (2018).
5. Osorio-Forero, A., Foustoukos, G., Cardis, R., Cherrad, N., Devenoges, C., Fernandez, L. M. J. & Lüthi, A. Locus coeruleus activity fluctuations set a non-reducible timeframe for mammalian NREM-REM sleep cycles. *bioRxiv* (2023) doi:10.1101/2023.05.20.541586.
6. Cataldi, J., Stephan, A. M., Marchi, N. A., Haba-Rubio, J. & Siclari, F. Abnormal timing of slow wave synchronization processes in non-rapid eye movement sleep parasomnias. *Sleep* **45**, (2022).
7. Pilon, M., Montplaisir, J. & Zadra, A. Precipitating factors of somnambulism symbol: impact of sleep deprivation and forced arousals. *Neurology* **70**, 2284–2290 (2008).
8. Mayer, G., Neissner, V., Schwarzmayr, P. & Meier-Ewert, K. Schlafentzug bei Somnambulismus. *Nervenarzt* **69**, 495–501 (1998).
9. Siclari, F., Baird, B., Perogamvros, L., Bernardi, G., LaRocque, J. J., Riedner, B., Boly, M., Postle, B. R. & Tononi, G. The neural correlates of dreaming. *Nat. Neurosci.*

- 20, 872–878 (2017).
10. Bernardi, G., Siclari, F., Yu, I., Zennig, C., Bellesi, M., Ricciardi, E., Cirelli, C., Ghilardi, M. F., Pietrini, P. & Tononi, G. Neural and behavioral correlates of extended training during sleep deprivation in humans: Evidence for local, task-specific effects. *J. Neurosci.* **35**, 4487–500 (2015).
 11. Mullen, T. R., Kothe, C. A. E., Chi, Y. M., Ojeda, A., Kerth, T., Makeig, S., Jung, T.-P. & Cauwenberghs, G. Real-Time Neuroimaging and Cognitive Monitoring Using Wearable Dry EEG. *IEEE Trans. Biomed. Eng.* **62**, 2553–2567 (2015).
 12. Plechawska-Wojcik, M., Kaczorowska, M. & Zapala, D. The Artifact Subspace Reconstruction (ASR) for EEG Signal Correction. A Comparative Study. in *Information Systems Architecture and Technology: Proceedings of 39th International Conference on Information Systems Architecture and Technology -- ISAT 2018* (eds. Świkatek, J., Borzemski, L. & Wilimowska, Z.) 125–135 (Springer International Publishing, 2019).
 13. Chang, C.-Y., Hsu, S.-H., Pion-Tonachini, L. & Jung, T.-P. Evaluation of Artifact Subspace Reconstruction for Automatic Artifact Components Removal in Multi-Channel EEG Recordings. *IEEE Trans. Biomed. Eng.* **67**, 1114–1121 (2020).
 14. Robbins, K. A., Touryan, J., Mullen, T., Kothe, C. & Bigdely-Shamlo, N. How Sensitive Are EEG Results to Preprocessing Methods: A Benchmarking Study. *IEEE Trans. Neural Syst. Rehabil. Eng.* **28**, 1081–1090 (2020).
 15. Anders, P., Müller, H., Skjæret-Maroni, N., Vereijken, B. & Baumeister, J. The influence of motor tasks and cut-off parameter selection on artifact subspace reconstruction in EEG recordings. *Med. Biol. Eng. Comput.* **58**, 2673–2683 (2020).
 16. Delorme, A. & Makeig, S. EEGLAB: an open source toolbox for analysis of single-trial EEG dynamics. *J. Neurosci. Methods* **134**, 9–21 (2004).
 17. Jung, T. P., Makeig, S., Humphries, C., Lee, T. W., Mckeown, M. J., Iragui, V. & Sejnowski, T. J. Removing electroencephalographic artifacts by blind source separation. *Psychophysiology* **37**, 163–78 (2000).
 18. Koch, C., Massimini, M., Boly, M. & Tononi, G. Neural correlates of consciousness: progress and problems. *Nat. Rev. Neurosci.* **17**, 307–321 (2016).
 19. Windt, J. M. *Dreaming: A Conceptual Framework for Philosophy of Mind and Empirical Research*. (MIT Press, 2015).
 20. Windt, J. M. Reporting dream experience: Why (not) to be skeptical about dream reports. *Front. Hum. Neurosci.* **7**, 708 (2013).
 21. Spiess, M., Bernardi, G., Kurth, S., Ringli, M., Wehrle, F. M., Jenni, O. G., Huber, R. & Siclari, F. How do children fall asleep? A high-density EEG study of slow waves in the transition from wake to sleep. *Neuroimage* **178**, 23–35 (2018).
 22. Bernardi, G., Betta, M., Cataldi, J., Leo, A., Haba-Rubio, J., Heinzer, R., Cirelli, C., Tononi, G., Pietrini, P., Ricciardi, E. & Siclari, F. Visual imagery and visual perception induce similar changes in occipital slow waves of sleep. *J. Neurophysiol.* **121**, 2140–2152 (2019).
 23. Osorio-Forero, A., Cardis, R., Vantomme, G., Guillaume-Gentil, A., Katsioudi, G., Devenoges, C., Fernandez, L. M. J. & Lüthi, A. Noradrenergic circuit control of non-REM sleep substates. *Curr. Biol.* **31**, 5009-5023.e7 (2021).
 24. Kjaerby, C., Andersen, M., Hauglund, N., Untiet, V., Dall, C., Sigurdsson, B., Ding, F., Feng, J., Li, Y., Weikop, P., Hirase, H. & Nedergaard, M. Memory-enhancing properties of sleep depend on the oscillatory amplitude of norepinephrine. *Nat. Neurosci.* **25**, 1059–1070 (2022).
 25. Broughton, R. J. Sleep disorders: Disorders of arousal? *Science (80-.)*. **159**, 1070–1078 (1968).

26. Parrino, L., Ferri, R., Bruni, O. & Terzano, M. G. Cyclic alternating pattern (CAP): The marker of sleep instability. *Sleep Med. Rev.* **16**, 27–45 (2012).
27. Lecci, S., Fernandez, L. M. J., Weber, F. D., Cardis, R., Chatton, J.-Y., Born, J. & Lüthi, A. Coordinated infraslow neural and cardiac oscillations mark fragility and offline periods in mammalian sleep. *Sci. Adv.* **3**, e1602026 (2017).
28. Bassetti, C., Vella, S., Donati, F., Wielepp, P. & Weder, B. SPECT during sleepwalking. *Lancet* **356**, 484–5 (2000).
29. Terzaghi, M., Sartori, I., Tassi, L., Rustioni, V., Proserpio, P., Lorusso, G., Manni, R. & Nobili, L. Dissociated local arousal states underlying essential clinical features of non-rapid eye movement arousal parasomnia: An intracerebral stereo-electroencephalographic study. *J. Sleep Res.* **21**, 502–506 (2012).
30. Seeber, M., Cantonas, L.-M., Hoevels, M., Sesia, T., Visser-Vandewalle, V. & Michel, C. M. Subcortical electrophysiological activity is detectable with high-density EEG source imaging. *Nat. Commun.* **10**, 753 (2019).
31. Fahimi Hnazaee, M., Wittevrongel, B., Khachatryan, E., Libert, A., Carrette, E., Dauwe, I., Meurs, A., Boon, P., Van Roost, D. & Van Hulle, M. M. Localization of deep brain activity with scalp and subdural EEG. *Neuroimage* **223**, 117344 (2020).

REVIEWERS' COMMENTS:

Reviewer #1 (Remarks to the Author):

I truly enjoyed reading this revised manuscript. The introduction and the goals of the study are now crystal-clear, and I appreciated the thorough and insightful responses of the authors to my comments. I don't have any additional comments, except to applaud the authors for this impressive study. I believe that this work will turn out to be very influential in the field. Congratulations!

Reviewer #2 (Remarks to the Author):

The authors have adequately addressed the majority of my concerns in their response. However, I still have a few remarks:

Abstract:

- In the abstract, I believe it is important to provide the percentage (%) along with the actual number of episodes related to the mentioned percentage (%) to give the reader a quick idea of the sample size.

Introduction

- The introduction appears more linear to me in this current version. Just a minor note: I apologize for not noticing this during the initial review, but for the sake of completeness, it might be beneficial to remind the reader - here (or in the discussion) - of the limited existing literature on EEG correlates in DOA. For example, the intracerebral EEG case by Sarasso et al. revealed an increase in beta activity at the thalamic level during a confusional arousal episode. Additionally, the intracerebral EEG case series by Flamand and the high-density case reports by Castelnovo et al. and Ratti et al. contribute to the understanding of this topic and other traditional EEG studies on the pre-episode period.

Discussion:

- In the discussion, I am slightly confused by the reasoning behind the slow wave findings that were added in the current version of the manuscript. In particular, I am not sure I fully grasped the statement starting at line 362: what does it mean by 'although in our previous study, it most consistently distinguished experiences with and without recall of content.'? Furthermore, as there is no clear classification of slow waves into Type I and Type II in the

current manuscript, I would be more careful in interpreting the results.

Methods:

- Line 131: Twenty of these participants were included in the previous work 54. > Probably, it is worth mentioning why these 2 subjects were excluded.
- I think that it should be mentioned somewhere, at least in the supplementary materials, that two patients had an AHI slightly above 15/h and two had a PLMS index of >15/h (23/h and 16.4/h). I also suggest stating how many patients had a diagnosis of OSAS (AHI > 5). Even more importantly, it might be interesting to mention how many episodes were triggered by respiratory or simple motor (PLM) events and if there was a difference between subgroups.
- Line 478: There is a contradiction between the statement: 'the end was defined as movement cessation' and the following one, added in the current version of the manuscript: 'In most cases, the end of an episode was marked by either a pause in movement, an interaction with the examiner to signal that patients were fully awake (see, for instance, hand clapping in video S2), or sometimes the resumption of a sleeping position'.
- I thank the authors for providing the paper on ASR validation during sleep, which I greatly appreciated. However, the paper clearly states: "the ASR cutoffs of 20-30 SD recommended for wake should never be used when studying the large graphoelements of non-REM sleep. As detailed later, we recommend using milder cutoffs of 30-45 SD in non-REM sleep to optimally remove artifacts while preserving SW amplitudes." If I understood correctly, the authors used a threshold of 25. What was the rationale for this choice? Was it based on the fact that the selected parasomnia episodes had visually lower slow waves compared to standard slow wave sleep? This is possible, as these are adult patients, but remains a critical point that should be clarified in my opinion.
- Line 525: Just out of curiosity: Were eye movements common during DOA episodes, in your experience, or did patients tend to have a 'stare' gaze with no or few eye movements?
- Line 523: This question possibly reflects my ignorance of the ASR method, but I still do not understand why keeping bad channels does not create trouble during the ASR calibration. Could you please provide a reference? If you used the "clean_rawdata" EEGLab plugin, could you also please state the other input parameters?
- Line 584: I am not entirely sure why it is necessary to add the 0.025 cut-off at this point, but I will leave the authors the final choice, as this does not interfere with data

interpretation.

- Indeed, there was no difference in the “time since lights off” between episodes with or without recall. Thanks for the clarification. Just one last question: why did the authors compute this variable from lights off and not from sleep onset?

Reviewer #3 (Remarks to the Author):

Cataldi et al have submitted an improved manuscript but the changes have not assuaged my concerns regarding:

1. The use of the Slow Wave metrics. The definition of the slow waves is based on arbitrary criteria and there is limited link to a biological mechanism.

See ' filtering (0.5-4Hz; stop-band at 0.1 and 10 Hz using a Chebyshev Type II filter), down sampling (from 500 to 128 Hz), duration threshold (only half-waves with a duration between 0.25s and 1s were considered), not including an amplitude threshold, and length of the moving average filter (50ms) for the derivative.'

And 'The precise functional and structural correlate of type I slow waves is still unknown'
So given the arbitrary nature and lack of biological significance, I am unsure what these analyses add, particularly over the delta power analyses.

2. Using multiple p-value thresholds does not give a sense of what was 'significant' in the binary frequentist approach. Please simplify and use one single threshold.

3. If source reconstruction is reliable and the parasomnia study is adequately powered why were there no differences in beta power in the CEWR vs. CE contrast? Conversely in the sleep dataset, there were no delta power differences between the CE and CEWR. While I admire the authors honesty and transparency, the data do not tell a clear story and do not seem to accurately underscore ' Amnesia for the experience (25%) was modulated by right hippocampal activation during prior sleep and fronto-parietal slow wave activity during the episode.'

4. While the study may be large for parasomnia research, the numbers of subjects (n=22) appears low to support the bold conclusions of the paper.

Responses to reviewers' comments.

Reviewer #1 (Remarks to the Author):

I truly enjoyed reading this revised manuscript. The introduction and the goals of the study are now crystal-clear, and I appreciated the thorough and insightful responses of the authors to my comments. I don't have any additional comments, except to applaud the authors for this impressive study. I believe that this work will turn out to be very influential in the field. Congratulations!

Reviewer #2 (Remarks to the Author):

The authors have adequately addressed the majority of my concerns in their response. However, I still have a few remarks:

Abstract:

- 1. In the abstract, I believe it is important to provide the percentage (%) along with the actual number of episodes related to the mentioned percentage (%) to give the reader a quick idea of the sample size.**

We understand the reviewer's concern, but the absolute number of parasomnia episode analyzed is not representative of the number of episodes recorded in this study, as only episodes with total agreement between the raters were selected. Therefore, we prefer not to include the absolute numbers in the abstract, so as not to mislead the reader on how many parasomnia episodes can be recorded with this type of study.

- 2. The introduction appears more linear to me in this current version. Just a minor note: I apologize for not noticing this during the initial review, but for the sake of completeness, it might be beneficial to remind the reader - here (or in the discussion) - of the limited existing literature on EEG correlates in DOA. For example, the intracerebral EEG case by Sarasso et al. revealed an increase in beta activity at the thalamic level during a confusional arousal episode. Additionally, the intracerebral EEG case series by Flamand and the high-density case reports by Castelnovo et al. and Ratti et al. contribute to the understanding of this topic and other traditional EEG studies on the pre-episode period.**

Thank you for pointing out the additional references. We have added the references of Flamand et al. and Sarasso et al. to the following sentence in the manuscript : "Previous studies using nuclear imaging methods and intracranial recordings documented patterns of wake- and sleep-like activity in different brain regions during NREM parasomnia episodes"¹⁻⁵, but these techniques are not readily available to study a large number of episodes."

And to referred to Ratti and Castelnovo in this part of the manuscript as follows:

Here we took advantage of high-density (hd-) EEG recordings to record parasomnia episodes, which we combined with a serial interview paradigm, a technique that has previously allowed us, in healthy sleepers, to document brain activity patterns that distinguish unconsciousness from dreaming in both REM and NREM sleep^{6,7}. Two previous case studies have used high-density EEG to image brain activity before and during episodes^{8,9}, demonstrating feasibility of the approach.

- 3. In the discussion, I am slightly confused by the reasoning behind the slow wave findings that were added in the current version of the manuscript. In particular, I am not sure I fully grasped the statement starting at line 362: what does it mean by 'although in our previous study, it most consistently distinguished experiences with and without recall of content.'?**

Furthermore, as there is no clear classification of slow waves into Type I and Type II in the current manuscript, I would be more careful in interpreting the results.

In our previous study, type I slow wave amplitude was highest in the DE condition, followed by the DEWR condition and the NE condition. Most consistent differences emerged between the DE and DEWR condition, while in the current manuscript, most consistent differences emerged between the DE and NE condition.

Although there is substantial evidence for two types of slow waves in both humans and rodents, their significance is not well understood. We have formulated our conclusions on type I slow waves more tentatively and eliminated the more speculative parts of the discussion.

- 4. Line 131: Twenty of these participants were included in the previous work 54. > Probably, it is worth mentioning why these 2 subjects were excluded.**

These two participants were not excluded, but instead acquired after the publication of the first study. We have added this information to the manuscript.

- 5. I think that it should be mentioned somewhere, at least in the supplementary materials, that two patients had an AHI slightly above 15/h and two had a PLMS index of >15/h (23/h and 16.4/h). I also suggest stating how many patients had a diagnosis of OSAS (AHI > 5). Even more importantly, it might be interesting to mention how many episodes were triggered by respiratory or simple motor (PLM) events and if there was a difference between subgroups.**

We have added the information on AHI/PLMS to the manuscript (supplementary information, paragraph 'additional information about patients'), as follows:

Two patients had a periodic leg movements of sleep (PLMS) index greater than 15/h (23/h and 16.4/h), and five patients had an AHI index greater than 5/h, of which two had an AHI greater than 15/h (16.7/h and 15.5/h).

We cannot determine whether the episodes we studied here were induced by PLM or OSAS events, because the experimental recordings (as opposed to the clinical PSG recordings) did not include respiratory or leg movement leads. However, we do not think that this should have in any way influenced our results.

- 6. Line 478: There is a contradiction between the statement: 'the end was defined as movement cessation' and the following one, added in the current version of the manuscript: 'In most cases, the end of an episode was marked by either a pause in movement, an interaction with the examiner to signal that patients were fully awake (see, for instance, hand clapping in video S2), or sometimes the resumption of a sleeping position'.**

We understand the apparent contradiction. The first sentence means that the EEG was always extracted until movement cessation. With the second sentence we describe what we usually saw in the video around the end of an episode. We realize the formulation 'was marked' is confusing in this respect, so we have rephrased as follows: In most cases, at the end of an episode patients paused, then sometimes started an interaction with the examiner to signal that they were fully awake (see, for instance, hand clapping in video S2), or resumed a sleeping position again.

- 7. I thank the authors for providing the paper on ASR validation during sleep, which I greatly appreciated. However, the paper clearly states: "the ASR cutoffs of 20-30 SD recommended for wake should never be used when studying the large graphoelements of non-REM sleep. As detailed later, we recommend using milder cutoffs of 30-45 SD in non-REM sleep to optimally remove artifacts while preserving SW amplitudes." If I understood correctly, the authors used a threshold of 25. What was the rationale for this choice? Was it based on the**

fact that the selected parasomnia episodes had visually lower slow waves compared to standard slow wave sleep? This is possible, as these are adult patients, but remains a critical point that should be clarified in my opinion.

The recommendations of the paper are valid for consolidated sleep (N2 and N3 sleep). In the present study, the ASR method was not applied to sleep but only to parasomnia episodes, which differed radically from consolidated sleep in terms of delta power/slow waves¹⁰. Therefore, the threshold for ASR was adapted to 25 SD after visual inspection of the parasomnia episode EEG data before and after the procedure, in order to efficiently remove the artifacts while conserving the slow waves.

- 8. Line 525: Just out of curiosity: Were eye movements common during DOA episodes, in your experience, or did patients tend to have a 'stare' gaze with no or few eye movements?**

Both patterns were observed: some patients stared ahead in a perplex manner, others presented more exploratory eye movements. Both type of eye movement patterns (staring and exploratory eye movements) could occur within the same episode.

- 9. Line 523: This question possibly reflects my ignorance of the ASR method, but I still do not understand why keeping bad channels does not create trouble during the ASR calibration. Could you please provide a reference? If you used the "clean_rawdata" EEGLab plugin, could you also please state the other input parameters?**

*The thresholds are defined using a robust metric, based on the distribution of power values for each PC component, in each short time window of the calibration data. If there are some bad channels it will therefore not affect the calibration seriously because they will be outliers. We used the "clean_rawdata" EEGLab plugin and we set the following parameters: sampling frequency, ASR threshold = 25, window length = 1.5*number of channel/sampling rate, use of GPU = false, maximum memory to use = 20*1024. The other parameters were set as default.*

- 10. Line 584: I am not entirely sure why it is necessary to add the 0.025 cut-off at this point, but I will leave the authors the final choice, as this does not interfere with data interpretation.**

We have redone the figures with one single threshold ($p < 0.05$).

- 11. Indeed, there was no difference in the "time since lights off" between episodes with or without recall. Thanks for the clarification. Just one last question: why did the authors compute this variable from lights off and not from sleep onset?**

We used 'time since light off' rather than 'time since sleep onset' in this analysis, as there were many 'sleep onsets' (after each parasomnia episode, or sometimes after induced awakenings with the sound), and only one 'time since lights off'. To address the reviewer's concern, we reran the analysis using 'time since the first sleep onset' instead, this analysis did not reveal an effect on consciousness or recall either. We have added this result to the manuscript and supplementary table 1.

Reviewer #3 (Remarks to the Author):

Cataldi et al have submitted an improved manuscript but the changes have not assuaged my concerns regarding:

- 12. The use of the Slow Wave metrics. The definition of the slow waves is based on arbitrary criteria and there is limited link to a biological mechanism. And 'The precise functional and structural correlate of type I slow waves is still unknown'. So given the arbitrary nature and**

lack of biological significance, I am unsure what these analyses add, particularly over the delta power analyses.

See ' filtering (0.5-4Hz; stop-band at 0.1 and 10 Hz using a Chebyshev Type II filter), down sampling (from 500 to 128 Hz), duration threshold (only half-waves with a duration between 0.25s and 1s were considered), not including an amplitude threshold, and length of the moving average filter (50ms) for the derivative.'

The filter and duration thresholds reflect the characteristic duration of sleep slow waves and have been used in many previous studies^{7,10-14}. Both intracranial and scalp recordings have shown that sleep slow waves recorded at the scalp level reflect oscillations of thalamo-cortical neurons at the same frequency^{12,15,16}, so there is a very robust neurobiological rationale for using these parameters. Although some studies include amplitude thresholds for slow waves, the selection of these amplitude thresholds is highly arbitrary. In the present study we decided not to include an amplitude threshold, so as not to introduce an a priori bias in favor of type I and type II slow waves. We saw that episodes with reported consciousness were preceded by smaller slow waves in posterior cortical regions and larger slow waves in anterior cortical regions. These conclusions are true regardless of the possible biological significance and even the existence of type I vs type II slow waves. We have further toned down the interpretation of the large slow waves as type I slow waves in the discussion.

13. Using multiple p-value thresholds does not give a sense of what was 'significant' in the binary frequentist approach. Please simplify and use one single threshold.

We have redone the figures with one single threshold ($p < 0.05$).

14. If source reconstruction is reliable and the parasomnia study is adequately powered why were there no differences in beta power in the CEWR vs. CE contrast? Conversely in the sleep dataset, there were no delta power differences between the CE and CEWR. While I admire the authors honesty and transparency, the data do not tell a clear story and do not seem to accurately underscore ' Amnesia for the experience (25%) was modulated by right hippocampal activation during prior sleep and fronto-parietal slow wave activity during the episode.'

Regarding the CEWR vs CE contrast in the sleep period, we did find higher beta power in the CE as opposed to the CEWR condition during sleep in the medial temporal region, along with decreased delta power, suggesting a sleep-related EEG activation of this area. During the episodes, CEWR episodes displayed higher delta power than CE, but no differences in beta power. Thus, the sentence 'amnesia for the experience (25%) was modulated by right hippocampal activation during prior sleep and fronto-parietal slow wave activity during the episode' adequately reflects our results. The absence of beta power differences between CE and CEWR during the episode (and also in the CE vs NE contrast), is not so surprising, since the parasomnia episodes are a very different behavioral state compared to sleep. As shown in our previous manuscript, there are huge changes in absolute power in the delta and beta frequency bands with respect to prior sleep (see¹⁰, Figure 1). In addition, contrary to prior sleep, during episodes participants move around and interact with their environment. Beta power reduction is commonly observed during movements^{17,18}, which could have influenced the findings. We have added this latter consideration to the discussion.

15. While the study may be large for parasomnia research, the numbers of subjects (n=22) appears low to support the bold conclusions of the paper.

We are confident that the main findings of the paper are statistically valid as they have been obtained using state-of-the-art mixed models, accounting for several confounds and were

corrected for multiple comparisons using an extremely strict procedure. In addition, the effect size for our previous study on dreaming with over 700 observations (experiment 2 of⁶) was large ($d = 1.4$). While large effect sizes can be found erroneously by chance in small studies, this does not happen by chance in a sample with over 700 observations. Based on this estimate, we were confident that assessment of a tractable number of patients would be sufficient to replicate the expected large effect of the main result, especially when using hypothesis-driven analyses. In this context we would like to stress the uniqueness of the data, which cannot be compared to more ordinary datasets obtained in sleep or wakefulness studies, as we are dealing with largely unpredictable events that are difficult to reproduce and capture in laboratory conditions. As reviewer 2 points out, only single cases or small case series on brain activity associated with parasomnia episodes, without assessment of consciousness, have been published so far, some of which have been hugely influential (see for instance⁴ on a single parasomnia episode). This is not to discredit previous work, but simply to put our present work into context.

References

1. Castelnovo, A., Lopez, R., Proserpio, P., Nobili, L. & Dauvilliers, Y. NREM sleep parasomnias as disorders of sleep-state dissociation. *Nat. Rev. Neurol.* **14**, 1–12 (2018).
2. Terzaghi, M. *et al.* Dissociated local arousal states underlying essential clinical features of non-rapid eye movement arousal parasomnia: An intracerebral stereo-electroencephalographic study. *J. Sleep Res.* **21**, 502–506 (2012).
3. Terzaghi, M. *et al.* Evidence of dissociated arousal states during nrem parasomnia from an intracerebral neurophysiological study. *Sleep* **32**, 409–12 (2009).
4. Bassetti, C., Vella, S., Donati, F., Wielepp, P. & Weder, B. SPECT during sleepwalking. *Lancet* **356**, 484–5 (2000).
5. Sarasso, S. *et al.* Fluid boundaries between wake and sleep: experimental evidence from Stereo-EEG recordings. *Arch. Ital. Biol.* **152**, 169–177 (2014).
6. Siclari, F. *et al.* The neural correlates of dreaming. *Nat. Neurosci.* **20**, 872–878 (2017).
7. Siclari, F., Bernardi, G., Cataldi, J. & Tononi, G. Dreaming in NREM sleep: A high-density EEG study of slow waves and spindles. *J. Neurosci.* **38**, 9175–9185 (2018).
8. Castelnovo, A. *et al.* High-density EEG power topography and connectivity during confusional arousal. *Cortex; a journal devoted to the study of the nervous system and behavior* **155**, 62–74 (2022).
9. Ratti, P.-L., Amato, N., David, O. & Manconi, M. A high-density polysomnographic picture of disorders of arousal. *Sleep* **41**, (2018).
10. Cataldi, J., Stephan, A. M., Marchi, N. A., Haba-Rubio, J. & Siclari, F. Abnormal timing of slow wave synchronization processes in non-rapid eye movement sleep parasomnias. *Sleep* **45**, (2022).
11. Riedner, B. A. *et al.* Sleep homeostasis and cortical synchronization: III. A high-density EEG study of sleep slow waves in humans. *Sleep* (2007). doi:10.1093/sleep/30.12.1643
12. Nir, Y. *et al.* Regional slow waves and spindles in human sleep. *Neuron* **70**, 153–69 (2011).
13. Bernardi, G., Siclari, F., Handjaras, G., Riedner, B. A. & Tononi, G. Local and widespread slow waves in stable NREM sleep: Evidence for distinct regulation mechanisms. *Front. Hum. Neurosci.* **12**, 1–13 (2018).
14. Siclari, F. *et al.* Two Distinct Synchronization Processes in the Transition to Sleep: A High-Density Electroencephalographic Study. *Sleep* **37**, 1621–1637 (2014).
15. Amzica, F. & Steriade, M. Electrophysiological correlates of sleep delta waves. *Electroencephalogr. Clin. Neurophysiol.* **107**, 69–83 (1998).
16. Steriade, M., Timofeev, I. & Grenier, F. Natural waking and sleep states: A view from inside neocortical neurons. *J. Neurophysiol.* **85**, 1969–85 (2001).
17. Zaepffel, M., Trachel, R., Kilavik, B. E. & Brochier, T. Modulations of EEG beta power during planning and execution of grasping movements. *PLoS One* **8**, e60060 (2013).
18. Toro, C. *et al.* Event-related desynchronization and movement-related cortical potentials on the ECoG and EEG. *Electroencephalogr. Clin. Neurophysiol.* **93**, 380–389 (1994).

REVIEWER COMMENTS

Reviewer #3 (Remarks to the Author):

Again the authors have not addressed my concerns.

Regarding slow waves, these analyses are another way of recapitulating the delta power results. Given the authors acknowledge that they lack biological significance, it is unclear what these analyses add. Removal of the slow wave analyses would simplify the paper. If the authors can show no relationship between size of the slow wave and delta power, I will concede these analyses add to the paper, if not please remove them.

Regarding the source space results, the authors are apparently "cherry picking" their interpretations. Either beta power in medial temporal regions is evidence of memory responses or not. The absence of beta power in these regions during a parasomnia is not consistent with this being a marker of memory. The authors' response exemplifies the difficulty in distilling the useful information in the paper and everyone's confusion as to what is the key information.

Overall, I think this work is valuable but is presently being oversold. It is essentially a simple paper but additional analyses make it all very complex to read. Part of this is the relative value of the sleep vs. parasomnia data. I would have thought it was the parasomnia EEG data that is most important and should be emphasized. Notably, with the memory data, I fail to understand how a lack of beta power responses in parasomnia leads to a conclusion that beta power is a marker of memory responses.

1. Again the authors have not addressed my concerns.

a. Regarding slow waves, these analyses are another way of recapitulating the delta power results.

The slow wave analysis is not equivalent to the spectral analysis. Rather, it breaks down the results of the spectral analysis, showing that the differences in posterior delta power are driven by a lower slow wave density in these regions in the consciousness condition compared to the unconsciousness condition. Crucially, this analysis shows that a few seconds before movement onset, slow wave amplitude is actually higher in the consciousness condition in frontal brain regions. This last finding is not apparent when analyzing only delta power, **as shown in the figure below**. This is because delta power reflects both slow wave amplitude, but just before movement onset, these two parameters are dissociated with respect to consciousness. Thus, the full constellation associated with consciousness consists in few slow waves in the back and big slow waves in the front of the brain, similar to what we previously found in our dream study. To recapitulate, the slow wave findings 1) support the conclusion that NREM dreams and parasomnia experiences share common regional patterns of brain activity and 2) also provide a potential explanation for why some reports are incoherent with behavior. Removing the important result on slow waves would certainly simplify the paper but would be equivalent to “hiding” that brain activity changes in frontal brain regions are associated with changes in consciousness. We cannot do this in good conscience. However, we understand that **these analyses may be difficult to understand** to a reader who is not from the field. We have therefore **moved the slow wave findings and related text in the discussion to the supplementary material**.

Figure. Top row: here the results of the delta power averaged over the last 20s are shown. Lower delta power in a posterior brain region predicts consciousness as opposed to unconsciousness (dotted region). **Middle row.** Here the results of delta power are broken down into two timeframes: -20 to -4s before movement onset and -4s to movement onset. Here we see that lower delta power in the sleep background (-20s to -4s) predicts consciousness, but not in the -4s to movement onset timeframe (corresponding to the arousal reaction leading up to the movement). The figures in the **lowest row** (slow wave analysis) show why: slow wave density and amplitude go in opposite directions with respect to consciousness. More specifically: the higher the amplitude slow waves in frontal regions and the lower the density in posterior regions, the more likely consciousness is.

- b. **Given the authors acknowledge that they lack biological significance, it is unclear what these analyses add.**

We have not stated that the K-complex-like slow waves lack biological significance, but that their function is not precisely known yet, similar to many other EEG patterns which are analyzed in sleep research. Although the function of high-amplitude anterior slow waves (K-complexes) is not precisely known, there is a **wealth of arguments to suggest that they are related to activation of arousal systems**, as outlined in the previous exchanges. This is an important aspect of the interpretation of the findings.

- c. **Removal of the slow wave analyses would simplify the paper. If the authors can show no relationship between size of the slow wave and delta power, I will concede these analyses add to the paper, if not please remove them.**

Indeed, as we show in the figure above, for some brain regions, there is an opposite relation between delta power and slow wave amplitude. As outlined in response to comment 1a, we are not comfortable hiding a result that likely reflects a key neurophysiological mechanism involved in sleep consciousness, even if its precise role is not yet understood. However, we are receptive to the argument that these analyses may be difficult to understand to a reader who is not from the field. Therefore, we have moved the slow wave results to the supplementary material of the paper, so that they remain accessible to the readers who are more familiar with this type of analysis. We have also moved part of the discussion on the slow wave findings to the supplementary material.

2. **Regarding the source space results, the authors are apparently "cherry picking" their interpretations. Either beta power in medial temporal regions is evidence of memory responses or not. The absence of beta power in these regions during a parasomnia is not consistent with this being a marker of memory. The authors' response exemplifies the difficulty in distilling the useful information in the paper and everyone's confusion as to what is the key information.**

As we have previously stated, we think that beta activity during parasomnia episodes cannot be interpreted in the same way than before movement onset because it is influenced by movement. In fact, we did not find beta power differences for any of the contrasts during the episode. We would also like to clarify that we never interpret beta findings on their own but together with delta, which goes in the opposite direction and together with the beta finding speaks for

hippocampal EEG activation. We have ***described the associative (rather than causal) relation with amnesia*** for the episode and ***have been cautious in our interpretation***: “right hippocampal deactivation during preceding sleep ... MAY prevent the encoding of conscious experiences (dreams) into episodic memory”. In addition, we have extensively discussed the limitations of the study in a dedicated paragraph. We are not sure which changes the reviewer would like to see in the manuscript in this regard. To further strengthen the associative (as opposed to causal) nature, we ***have rephrased the findings on amnesia*** in the abstract.

- 3. Overall, I think this work is valuable but is presently being oversold. It is essentially a simple paper but additional analyses make it all very complex to read. Part of this is the relative value of the sleep vs. parasomnia data. I would have thought it was the parasomnia EEG data that is most important and should be emphasized. Notably, with the memory data, I fail to understand how a lack of beta power responses in parasomnia leads to a conclusion that beta power is a marker of memory responses.**

We are glad to hear that the reviewer thinks the work is valuable. We hope that by moving the slow wave analyses and some text to the supplementary material, the paper has become easier to read. We agree that the EEG after movement onset is highly interesting. However, its interpretation should take into account that 1) it is not the same behavioral state, the EEG being radically different from sleep, as we have previously shown (Catladi et al, Sleep 2022) and 2) participants moved and interacted with the environment this state with eyes open. To simplify, we used the term “sleep” to refer to brain activity before movement onset (20s), and ‘episode’ to refer to the time window ranging from +4s to +20s, but it should be clear that movement onset is necessarily preceded by brain activity changes that lead up to it, in other words, movement onset is the first visible externalized event of an arousal process that necessarily starts before. Indeed, as the slow wave analysis over finer time bins shows, the last 5s before movement onset already show brain activity that is different from the sleep background. This means that ***brain activity changes that happen immediately before movement onset already reflect the parasomnia process. For clarity, we have reformulated the terminology in this respect.***

REVIEWERS' COMMENTS

Reviewer #3 (Remarks to the Author):

The paper is improved for being simpler. A point of confusion for me is that argument for divergent effects of high beta and low delta being critical for activation but not analysis of the ratio. Nonetheless, the arguments are simpler in the present presentation and while incremental over Siclari et al Nat Neuro they are useful addition to the literature.

Thank you for explaining the regional differences in SW density and amplitude and delta power. I think topographic correlations of these factors would be useful in the paper (as the data presented based on T thresholds cannot be used to argue a lack of relationship of the variables). This is important as frontal SW have larger amplitude and frontal delta power is higher than more posterior regions.

Please add the caveat these data only show associations and there can be no causal inference based on these findings.

The paper is improved for being simpler. A point of confusion for me is that argument for divergent effects of high beta and low delta being critical for activation but not analysis of the ratio. Nonetheless, the arguments are simpler in the present presentation and while incremental over Siclari et al Nat Neuro they are useful addition to the literature. Thank you for explaining the regional differences in SW density and amplitude and delta power. I think topographic correlations of these factors would be useful in the paper (as the data presented based on T thresholds cannot be used to argue a lack of relationship of the variables). This is important as frontal SW have larger amplitude and frontal delta power is higher than more posterior regions.

We agree with the reviewer that representations of t-values is not fully informative in this regard. For this reason, in addition to topographic representation of t-values, we have provided maps of absolute spectral power and all slow wave parameters for the CE and NE conditions separately in the supplementary material (Fig S1+S4). As requested from the reviewer, below we provide topographic correlations between delta power and slow wave parameters. Both slow wave amplitude and density are positively correlated with delta power, although slow wave amplitude correlates more strongly and consistently across channels. Of note, this result does not mean that slow wave amplitude and density cannot dissociate, which they do in the CE vs NE conditions, as shown in supplementary Figures S1 and S4 and in the figures provided in response to the last round of revisions. We are not convinced the correlations provided below are informative for the scope of the paper, and, given the abundance of information in the supplementary material, would prefer not to include them.

Correlations between delta power (1-4 Hz) and slow wave parameters

Figure. Topographic representation of Spearman correlation coefficients across all trials of sleep preceding parasomnia episodes (-20s to movement onset). White dots denote channels for which the correlation was statistically significant ($p < 0.05$, uncorrected). Similar results were found when analyzing NE and CE trials separately.

Please add the caveat these data only show associations and there can be no causal inference based on these findings.

We have added this statement as follows to the limitation paragraph: Finally, most of our results are based on associations, not allowing for causal inference.